# On the attribution of industrial-era glacier mass loss to anthropogenic climate change

Gerard H. Roe[1], John Erich Christian[1], Ben Marzeion[2]

[1] Dept. Of Earth and Space Sciences, U. Washington, Seattle, WA, USA
[2] Institute of Geography and Centre for Marine Environmental Sciences, U. Bremen, Germany

*Correspondence to*: Gerard H. Roe (groe@uw.edu)

**Abstract.** Around the world, small ice caps and glaciers have been losing mass and retreating during the industrial era. Estimates are that this has contributed approximately 30% of the observed sea-level rise over the same period. It is important to understand the relative importance of natural and anthropogenic components of this mass loss. One recent study concluded that the best estimate of the magnitude of the anthropogenic mass loss over the industrial era was only 25% of the total, implying a predominantly natural cause. Here we show that the fraction of the anthropogenic mass loss to the total mass loss of a given glacier depends only on the magnitudes and rates of the natural and anthropogenic components of climate change, and on the glacier's response time. We consider climate change over the past millennium using synthetic scenarios, paleoclimate reconstructions, numerical climate simulations, and instrumental observations. We use these climate histories to drive a glacier model that can represent a wide range of glacier response times to evaluate the magnitude of the anthropogenic mass loss relative to the observed mass loss. The slow cooling over the preceding millennium, followed by the rapid anthropogenic warming of the industrial era means that, over the full range of response times for small ice caps and glaciers, the central estimate of the magnitude of the anthropogenic mass loss is essentially 100% of the observed mass loss. The anthropogenic magnitude may exceed 100% in the event that, absent anthropogenic climate forcing, glaciers would otherwise have been gaining mass. Our results bring assessments of attribution of glacier mass loss into alignment with assessments of others aspects of climate change, such as global-mean temperature. Furthermore, these results reinforce the scientific and public understanding of centennial-scale glacier retreat as an unambiguous consequence of human activity.

## 1 Introduction

Over the past one hundred and fifty years, mountain glaciers and ice caps around the world have retreated dramatically, and associated with this retreat has been a loss of ice mass. Images of glacier retreat are an iconic part of the public communication of anthropogenic climate change; and the loss of ice mass has contributed to global sea-level rise (Zemp et al., 2019), hazards in deglaciating alpine environments (Stuart-Smith et al., in review), and downstream impacts on river-flow and water security (Milner et al., 2017). It is important to evaluate the anthropogenic component of these changes.

The process of characterizing the anthropogenic contribution to glacier mass loss is an exercise in detection and attribution (e.g., Bindoff et al., 2013). In terms of detection, the signal is clear. For instance, Zemp et al. (2015) analyse the full observational record of length and mass-balance: outside of Antarctica, glaciers are all smaller than they were in 1850, and the rate of mass loss in the early twenty-first century is unprecedented. In terms of attribution to a specific cause, Roe et al. (2017) demonstrated that the observed temperature trends since 1880 have caused glacier retreat that far exceeds the length fluctuations that would occur due to natural climate variability. Other studies have established that those temperature trends are primarily driven by anthropogenic forcing (e.g., Allen et al., 2018; Haustein et al., 2019). In combination with these other studies, Roe et al. (2017) constitutes a statement of attribution on glacier retreat.

Another approach is to use global climate models to simulate the impact of all known climate forcings on glacier mass balance, and compare with so-called 'counterfactual' simulations, in which the anthropogenic forcings are omitted (e.g., Hirabayashi et al., 2016; Vargo et al., 2020). The difference is the anthropogenic effect. For instance, Hirabayashi et al. (2016) apply this approach to the mass balance of eighty-five glaciers, and report a detectable anthropogenic influence consistent with observations.

One limitation of the studies noted above is that they involve only well-observed glaciers, which are a small subset of the total. This does not necessarily weaken the strength of any attribution statements, since the global picture can be abundantly clear without requiring an individual assessment for every glacier. However, for the purpose of assessing the anthropogenic contribution to sea-level rise, one must try to estimate human influence on glaciers in the global aggregate. To this end, Marzeion et al. (2014, hereafter M14) uses output from pairs of climate-model ensembles -- one ensemble with all-known climate forcing, and one with just natural forcing. The coarse output from the climate model is downscaled to the full global dataset of ~180,000 glaciers (Pfeffer et al., 2014, not including Antarctica and the main Greenland ice sheet), via a simplified mass-balance scheme based on precipitation and temperature. M14 concludes that between 1850 and 2010, and in the global average, only 25% ($\pm$35% at 1$\sigma$) of the mass loss was due to anthropogenic emissions (greenhouse gases and aerosols); and it rises to 69% ($\pm$24%) for the interval 1990 to 2010. These results imply that anthropogenic forcing was not the predominant cause of the observed mass loss since 1850. And further, if the results were assumed to apply to all glaciers, anthropogenic

forcing would also not account for the observed glacier retreat even over the 1990 to 2010 interval, because glacier retreat lags glacier mass loss by a decade or longer. Such a result might then call into question the use of images of glacier retreat as exemplary of the impacts of climate change.

A globally comprehensive assessment of glacier-mass loss is an important task, but it is one that pushes the limits of the
available information. In the face of a warming climate, glaciers are in a state of disequilibrium -- they are playing catch-up with the evolving climate. This state of disequilibrium must be estimated to correctly initialize each glacier simulation. The transient evolution of the glacier is controlled by the glacier response time, as well as by the shape of the response function associated with the ice dynamics. These too must be estimated. Finally, many numerical simulations of climate change are initialized at the onset of significant anthropogenic emissions, meaning mid-to-late nineteenth century. This leads to
uncertainty about whether prior climate history may have left glaciers in a state of substantial disequilibrium at the start of the industrial era, which would affect the subsequent evolution of glacier length and mass balance.

In this study, we re-evaluate the attribution of glacier-mass loss to anthropogenic forcing. We consider climate change over the whole past millennium, which helps avoid issues associated with uncertain initial conditions of the glaciers at the beginning
of the industrial era, and we explicitly consider a wide range of glacier response times. To illustrate the physical principles involved, we create synthetic representations of natural and anthropogenic climate change. We also use millennial-scale reconstructions of summertime temperature from proxy data, millennial-scale counterfactual climate modelling that has recently become available, and instrumental observations. These climate time series are then used as input to a glacier model from which the mass balance can be calculated for a wide range of glacier response times from 10 to 400 years. All our analyses
indicate a much larger role for anthropogenic forcing than estimated by M14. For all climate scenarios considered, and for glaciers with response times up to several centuries, we conclude that anthropogenic forcing has been the predominant driver of glacier mass loss since 1850.

## 2 Physical principles

For simplicity, consider a climate with no interannual variability. In a constant climate, a glacier will have a constant length
and its net mass balance will be zero. Figure 1 illustrates the basic response of glacier length and mass balance for three simple climate scenarios, and for two different glacier response times ($\equiv \tau$). For the purpose of illustration, we've picked $\tau=20$ yrs (red lines) and $\tau=60$ yrs (orange dashed lines).

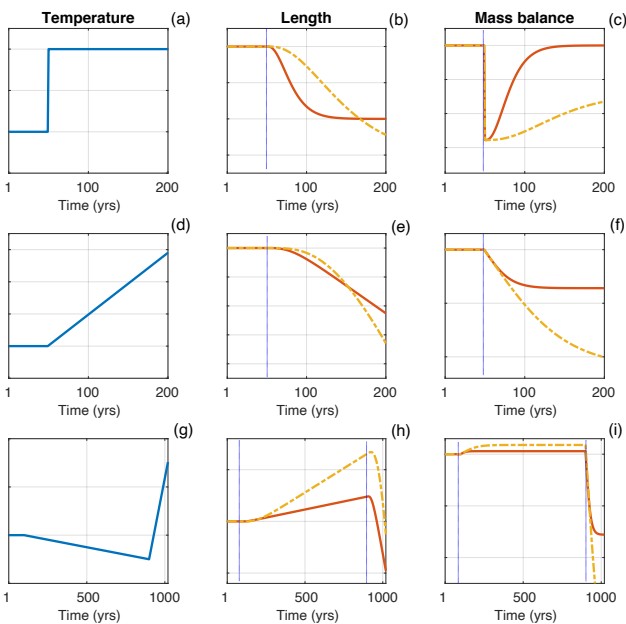

**Figure 1. An illustration of three simple climate scenarios (left panels), and the response of glacier length (centre panels) and mass balance (right panels), for two glacier response times: $\tau=20$ yrs (red lines), and $\tau=60$ yrs (orange dashed line). The magnitudes of the changes are essentially arbitrary in these examples, but serve to illustrate the basic principles of the response.**

First, consider a step-change increase in temperature (Fig. 1a). Both glaciers retreat, and their lengths asymptote towards a new equilibrium on a timescale set by glacier's dynamical response (Fig. 1b). The shorter-$\tau$ glacier reaches a new equilibrium state within the timeframe illustrated, but the longer-$\tau$ glacier has yet to get there, even 150 years after the anomaly. At the moment of the step change, both glaciers are in immediate negative mass balance. This imbalance recovers back towards zero as the glaciers' geometry adjusts (Fig. 1c).

Second, consider a linear warming trend (Fig. 1d). The glaciers' length responses to the temperature change are delayed, and by an amount that is again governed by their response times. Focusing first on the shorter-$\tau$ glacier, it takes time to spin up but eventually acquires a linear retreat rate with a constant lag (Fig. 1e). In this state, the glacier is always playing catch up to the warming climate, and the resulting disequilibrium is reflected in a mass-balance deficit that evolves to a constant value (Fig. 1f). The longer-$\tau$ glacier follows a similar trajectory, but lags further behind the linear trend, and has not fully spun up to the applied trend over this timeframe.

Third, consider a millennium-scale scenario of a slow cooling followed by a rapid warming (Fig. 1g). During the cooling, the glaciers advance (Fig. 1h) and are in a state of positive mass balance (Fig. 1i). Because the cooling is slow relative to the response times, the glaciers are in near-equilibrium with the climate, and the mass balance is only slightly positive. However, the longer-$\tau$ glacier lags further behind the cooling and so has a more-positive mass balance (Fig. 1i). After the switch to a

115 rapid warming trend, the glaciers may continue to grow for a short period as positive thickness anomalies continue to push the termini forward (e.g., Roe and Baker, 2014). However, the mass balance declines immediately, and becomes negative shortly before the glaciers reach their maximum length. The rate of warming is greater than the rate of cooling, and so the glaciers are in a greater state of disequilibrium during the period of warming: in terms of absolute magnitude, the rate of mass loss during the rapid warming period is much greater than the rate of mass gain during the earlier slow cooling period (Fig. 1i). If we

associate the rapid warming in Fig. 1g with industrial-era anthropogenic climate change, then Fig. 1i shows that the cumulative mass loss over that period (i.e., the areas under the curves in Fig. 1i) depends on each glacier's response time and initial state at the start of industrial era.

To summarize the principles illustrated in Fig. 1, the overall trajectories of glacier length and mass balance depend on the glacier response time, and the magnitudes and rates of the different components of the climate change.

## 3 A model of glacier response

Roe and Baker (2014) showed that the ice dynamics of alpine glaciers can be well emulated by a linear third-order equation:

$$\left(\frac{d}{dt} + \frac{1}{\epsilon\tau}\right)^3 L' = \frac{1}{\epsilon^3\tau^2}\beta b_f', \qquad (1)$$

where $L'(t)$ is the length anomaly from some defined equilibrium state, $\bar{L}$ ; $\tau$ is the glacier response time (see next section); $\beta$ is a geometric constant that affects glacier sensitivity; and $\epsilon = 1/\sqrt{3}$. $b_f'$ is the anomalous mass-balance forcing with respect to the equilibrium geometry, or equivalently, a reference surface (e.g., Huss et al., 2012). That is, $b_f'$ is not affected by the glacier's geometric adjustment; Equation (1) reflects three overlapping dynamical stages of glacier adjustment (Roe and Baker, 2014; Hooke, 2019), and accurately reproduces the transient length response of numerical flowline models, for both the shallow-ice approximation and full-Stokes ice dynamics (Christian et al., 2018).

Equation (1) is linear, which allows the effects of different climate forcing to be disaggregated into components. Note that $\beta$ is a constant for each glacier. Therefore, when we evaluate the anthropogenic mass loss relative to the total mass loss, $\beta$ cancels out when the ratio is taken. This useful result means that the detailed geometry of a glacier does not matter, except to the degree that it affects $\tau$.

### 3.1 Glacier response time.

For alpine glaciers the response time is governed by the ice thickness, $H$, and net (negative) mass balance near the terminus, $b_t$ (Jóhannesson et al., 1989):

$$\tau = -\frac{H}{b_t} \qquad (2)$$

Roe and Baker (2014) showed that this timescale governs the dynamics of a wide range of sizes of alpine glaciers. Jóhannesson at al. (1989), and much of the literature that has followed, has treated this is as an e-folding timescale. However, glacier length does not approach a new equilibrium as a simple exponential function – the shape of the curve is sigmoidal (e.g., Fig. 1b, or Roe and Baker, 2014). As a result, the time taken to approach to $1/e$ of the new equilibrium is approximately $1.8\tau$ (Roe and Baker, 2014). Estimating or assuming exponential responses times can lead to significant errors in assessing a glacier's sensitivity to climate and its state of disequilibrium (Christian et al., 2018).

$\tau$ is a parameter whose value can be selected, so a wide range of values can be evaluated. We consider five different glacier response times: $\tau = 10, 30, 100, 200,$ and 400 yrs. At the low end of that range are glaciers with steep terminal slopes and/or high terminus ablation rates (Nisqually glacier, U.S., ~10 yrs; Franz Josef glacier, N.Z., ~10 yrs, e.g., Roe et al., 2017). At the high end of our range, outlet glaciers from the larger ice caps and fields (outside of Greenland and Antarctica) may get up to such values, with ice thickness often in excess of 500 m (e.g., Koerner, 1977). Mass balance rates and gradients are low in the

high arctic, and $b_t$ of ~0.5 to 3 m yr$^{-1}$ appear typical (e.g., Hagen et al., 2003; Burgess and Sharp, 2004). Burgess and Sharp (2004) estimate response times of 125 yrs to nearly 1000 yrs for outlet glaciers of the Devon ice cap. Multi-century response times (Eq. 1) are thus a reasonable upper bound for our analyses.

For context, using the Jóhannesson et al. (1989) formula and some geometric approximations, Haeberli and Hoelzle (1995)
give a range for $\tau$ of 15 to 125 yrs for a population of 1800 glaciers in the Alps; Christian et al. (*in review*) give a range of 10 to 100 yrs for 380 glaciers in the Cascades (with 90% between 10 and 60 yrs); and Roe et al. (2017) give a range of 10 to 90 yrs for 39 glaciers around the world. Harrison et al. (2001) modified Jóhannesson et al. (1989) by noting that particularly low-sloped glaciers can have larger $\tau$ because of a mass-balance feedback. Lüthi et al. (2010) used this modified approach and found a range from 16 to 146 yrs for 13 glaciers in the Alps. To further explore the upper end of the range of $\tau$ for larger
glaciers, we applied the Haeberli and Hoelzle (1995) scaling to the Arctic Canada South and North regions of the Randolph Glacier Inventory (Pfeffer et al., 2014), assuming mass balance gradients typical of the region (~ 0.2 m yr$^{-1}$ per 100 m; e.g., Hagen et al., 2003; Burgess and Sharp, 2004). For the approximately 5000 glaciers that were larger than 1 km$^2$ and have vertical extent more than 250 m, we found that 97% by number, and 90% when weighted by area, had $\tau$ less than 400 years (not shown). This suggests that 400 years is a reasonable upper bound to take for our analyses. There are a variety of other
approaches for estimating $\tau$, many of which assume an exponential fit to time series of numerical models or data (e.g., Leclerq et al., 2012; Zekollari et al., 2020). We believe that the range of $\tau$ we consider here encompasses the overwhelming majority of alpine glaciers and small ice caps. The limiting cases are worthwhile to keep in mind: in the limit $\tau \to 0$, the mass balance is always zero, and the glacier length reflects the instantaneous climate. In the limit $\tau \to \infty$, the glacier is a static ice field, and it is the mass balance that reflects the instantaneous climate.

### 3.2 Glacier mass balance.

Let us define $b'_{eq}(t) = L'(t)/\beta\tau$. From Eq. (2), this is the mass-balance perturbation that is in equilibrium with the glacier length, $L'(t)$. The mass balance that would be measured on the glacier at any given time (i.e., the conventional mass balance, e.g., Huss et al., 2012) is then given by $b' = b'_f - b'_{eq}$, which includes both the climate forcing and the evolving glacier length.
The total rate of mass loss from the glacier is then $b'(\overline{L} + L'(t))$. Consistent with the linear approximation, we drop the quadratic term, leaving $b'\overline{L}$. Thus, the total rate of mass loss scales with $b'$ multiplied by a constant, albeit one that varies for different glaciers. We evaluate the validity of this linear approximation (supplementary material, $\equiv$ SM, Sec$^n$. S2). A correction

to the absolute mass loss is needed for small glaciers that have nearly disappeared over the industrial era (i.e., $L' \sim \bar{L}$), but for glaciers with large remaining areas, it has only a small impact on the fraction of the loss that is attributable to anthropogenic activity.

## 4 Analysis

In this section we consider synthetic, reconstructed, modelled, and instrumental records of climate history. We use the time series of these histories to drive the Roe and Baker (2014) glacier model over a wide range of glacier response times, and we evaluate the fraction of glacier mass loss that is modelled to be, or can be inferred to be, due to anthropogenic activity.

### 4.1 Synthetic Little-Ice-Age-like last millennium.

We first create a synthetic representation of the climate of the so-called Little Ice Age (LIA) climate of the last millennium. Temperature anomalies are related to mass-balance anomalies using a melt factor $\mu = 0.65$ m yr$^{-1}$ K$^{-1}$ (e.g., Herla et al., 2017). We impose a slow natural cooling of 0.25 °C kyr$^{-1}$ over the whole millennium as indicated by many proxy-based temperature reconstructions, and also by global climate models forced with only natural factors (e.g., Otto-Bliesner et al., 2016). Then starting in 1880, we add an anthropogenic warming of 1.0 °C per century up to the year 2020. On top of these long-term changes we also add interannual variability in the form of white noise with a standard deviation of 0.5 °C (Fig. 2a). This produces a signal-to-noise ratio in the anthropogenic mass-balance forcing of two, which is comparable to many locations around the world (Roe et al., 2017). In these idealized scenarios, we omit precipitation forcing of mass balance. Precipitation is considered in Sections 4.3 and 4.4. Since trends in precipitation associated with climate change are generally weak (e.g., Hartmann, 2013; Parsons et al., 2017), the main effect of including precipitation is to add extra interannual variability in mass balance. This is discussed further in Section S5 of the supplementary material (SM).

The panels in Fig. 2 show the glacier length, mass balance, and anthropogenic component for our five different values of $\tau$ over the last millennium. Figure 3 shows the same information, but for clarity and comparison with M14, it zooms into the period 1850 to 2020. We emulate the approach of M14 by showing the response to just the LIA cooling (which is the natural-forcing scenario, $\equiv$ NAT), and then also the combined response that includes the anthropogenic warming ($\equiv$ FULL, for full-forcing case). Thus, the magnitude of the anthropogenic mass loss is ANTH = FULL - NAT. Note that FULL is the model equivalent of the observed rate of mass loss in Nature.

We here take a moment to discuss various possible metrics that can be used to characterize the influence of anthropogenic factors on glacier mass balance. Each have different merits. One approach is to set up a null hypothesis of a counterfactual, purely natural climate with no anthropogenic forcing, and to pose the question "how likely is the observed change in the natural

case?". If the likelihood of the observed change falls below some stated level of significance, then the null hypothesis is rejected, and a "signal" can be declared detected. This was the approach applied to glacial retreat in Roe et al. (2017), and we apply it to mass balance in Section 4.4 of this paper. Another approach is to define a measure of the variance explained among an ensemble of climate models where natural variability, different anthropogenic forcing scenarios, and different models all contribute to the spread (e.g., Hawkins and Sutton, 2009; Marzeion et al., 2020), which is helpful for partitioning uncertainty. A third approach, one that has been used by the Intergovernmental Panel on Climate Change (IPCC) over several past reports, is to estimate the magnitude of the anthropogenic warming relative to the observed change. For instance, as of the 2018 IPCC Special Report, the central estimate of the magnitude of the anthropogenic change in global-mean surface temperature over the industrial era is that it is equal to 100% of the observed warming, with an assessed *likely* range (i.e., a 2-in-3 chance) of $\pm 20\%$ (Allen et al., 2018). Note this uncertainty range allows for the possibility that the magnitude of the anthropogenic warming exceeds 100% of the observed, because it is possible that the natural climate would have otherwise been cooling. M14 followed the IPCC in using this approach. It is useful, for instance, in the context of interpreting sea-level rise, where other contributions from thermal expansion and large ice sheets need to be understood. We want our results here to parallel those of M14, and we adopt the same approach as M14 in Sections 4.1 and 4.3 of this study.

M14 defined the following metric for comparing the magnitude of the anthropogenic mass loss to the FULL mass balance:
$F_{anth} = 100\% \times (ANTH/FULL) = 100\% \times (1 - NAT/FULL)$. M14 calculated the 20-yr running mean of $F_{anth}$, in order to assess how anthropogenic mass loss has evolved over time at decadal scales. For assessing the total change in mass over the industrial era, *ANTH* and *FULL* can be integrated over time before calculating their ratio. Note that this cumulative ratio will not, in general, be equal to the time-average of $F_{anth}$. In our analyses, we present both the decadal and cumulative values. Both provide useful information: the 20-yr running mean provides insight into the causes of decadal-scale mass-balance observations, whereas the cumulative anthropogenic mass loss can be more directly tied to glacier retreat (see Section 4.4) and sea-level rise.

Note also that $F_{anth}$ can behave a little strangely, especially early in the industrial era, because the mass balance can change sign. If ANTH has the opposite sign to FULL, then $F_{anth}$ can be negative. Moreover, $F_{anth}$ will be infinite as FULL crosses zero. As already noted, $F_{anth}$ can also be larger than 100% if, for instance, the glacier would have been growing in the NAT case. So, particularly right after the onset of the anthropogenic trend, $F_{anth}$ can be jumpy. It becomes better behaved as the magnitude of the anthropogenic component becomes large. The magnitude of the cumulative anthropogenic mass loss relative to the cumulative total is displayed as a percentage in the lower right of the right-hand panels of Figs. 3, 5, and 8.

Turning back to Figs. 2 and 3, during the slow cooling between 1000 and 1880 the glaciers grow, with interannual variability causing some fluctuations around the long-term trend. For all values of $\tau$ , the long-term mass balance is positive over this interval, but it is only slightly so because the glaciers are in near-equilibrium with the long-term cooling. Interannual variability

causes rapid fluctuations in mass balance that straddle this small disequilibrium and routinely cross zero (note the 20-yr running mean applied to the mass balance time series in Figs. 2 and 3). At the onset of the anthropogenic trend, the $\tau = 10$ yr glacier

adjusts rapidly, and quickly attains a state of small negative mass balance (Fig. 3c cf. Fig. 1f). Using the M14 definition, $F_{anth}$ initially spikes negative and then goes through a singularity as the FULL mass balance goes through zero. $F_{anth}$ then remains above 100% because, for this synthetic history, glaciers would have been growing in the absence of anthropogenic forcing. When interannual variability is included, $F_{anth}$ is quite jumpy: in the FULL case, the relatively small long-term negative mass balance means that there are intervals where natural variability makes the mass balance positive. Note that, for small-enough

$\tau$, these intervals of positive mass balance can cause temporary glacier advances against the backdrop of overall retreat (Fig. 3b). However, for the long-term average, and over the whole period (or very quickly after 1880) $F_{anth}$ exceeds 100%.

For glaciers with longer $\tau$, the adjustment at the onset of the anthropogenic trend is slower; the glacier length lags further behind the warming climate; and so the state of mass-balance deficit is larger. Only the $\tau = 10$ yr and $\tau = 30$ yr cases fully

spin up to the applied trend by 2020. For longer $\tau$, the larger long-term deficit in mass balance means the $F_{anth}$ curves are less jumpy. However, for all $\tau$, $F_{anth}$ rapidly rises above 100% within a few years of the onset of the trend, and it stays there. The percentages displayed in the right-hand panels of Fig. 3 compare the magnitudes of the cumulative anthropogenic mass loss to the cumulative total mass loss (i.e., ANTH/FULL, where both numerator and denominator are the integrals of the mass balance from 1880 to 2020). For all $\tau$, this ratio exceeds 100%.


Note that, by the end of the period shown, the length of the larger-$\tau$ glaciers have only just started to respond to the warming trend. The $\tau = 400$ yr glacier has not yet begun to retreat, even though it is losing mass: thickness anomalies built up during the previous cooling continue to push the glacier forwards. Explained in terms of the glacier's spectral response, the multi-stage process of glacier adjustment means the glacier length can lag the climate forcing by more than 90° (i.e., the phase angle

exceeds quadrature, Lüthi, 2009; Roe and Baker, 2014).

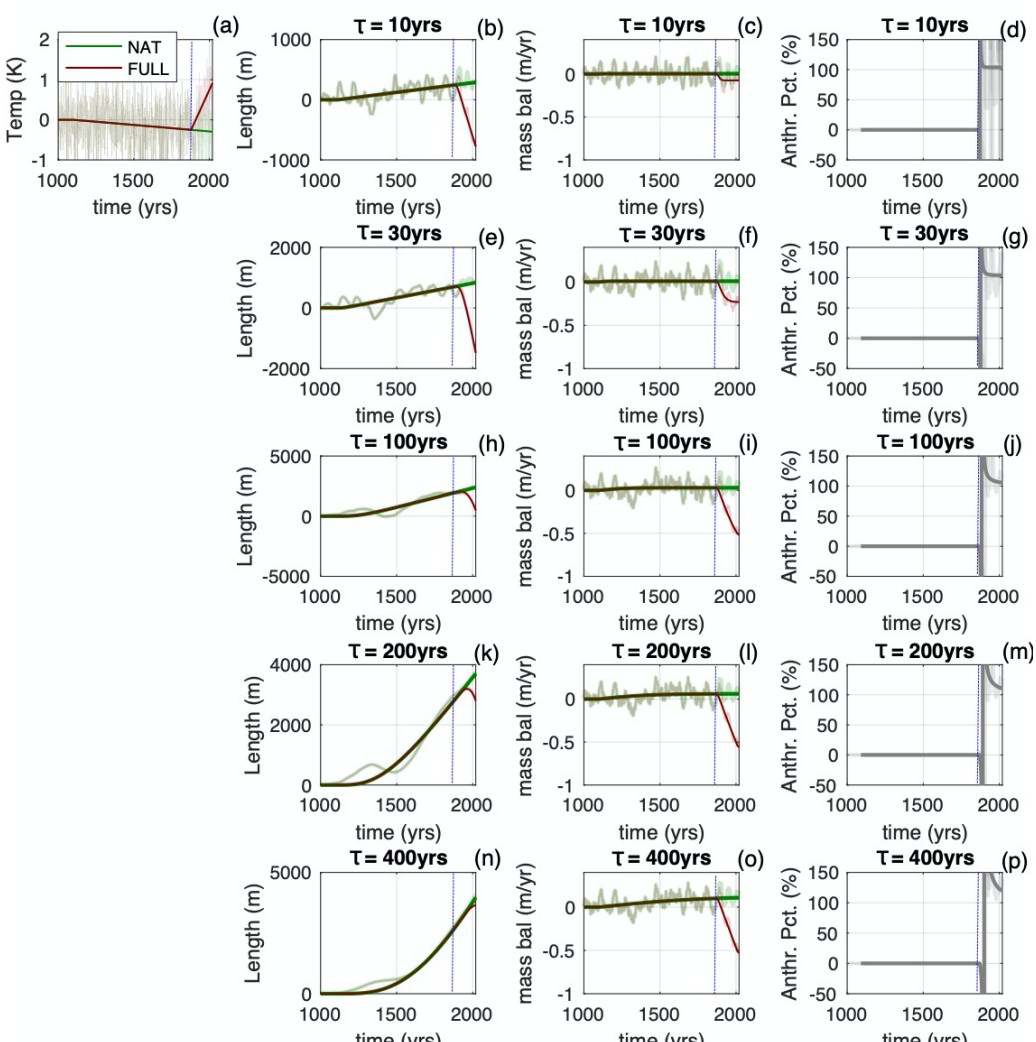

**Figure 2. Synthetic simulations of the last millennium. (a) temperature history, with 'NAT' case (green line), and 'FULL' case (red line). The lighter thinner lines have white noise added. The columns on the right show, in each row, the length, the mass balance, and the magnitude of the anthropogenic mass loss relative to the total (i.e., $F_{anth}$) for glaciers with timescales of 10, 30, 100, 200, and 400 yrs, respectively. The mass balance and $F_{anth}$ have a 20-yr running mean applied, as in M14.**

The underlying reasons for $F_{anth}$ are different for different glacier response times. In the short-$\tau$ limit, the glacier has little memory of prior climate history, and so it is dominated by whatever is happening recently. In the long-$\tau$ limit, the glacier's

length changes so slowly that its mass balance directly reflects the current climate. In between those limits, there is a blend of the inherited climate history being progressively forgotten and the ice dynamics trying to adjust towards equilibrium.

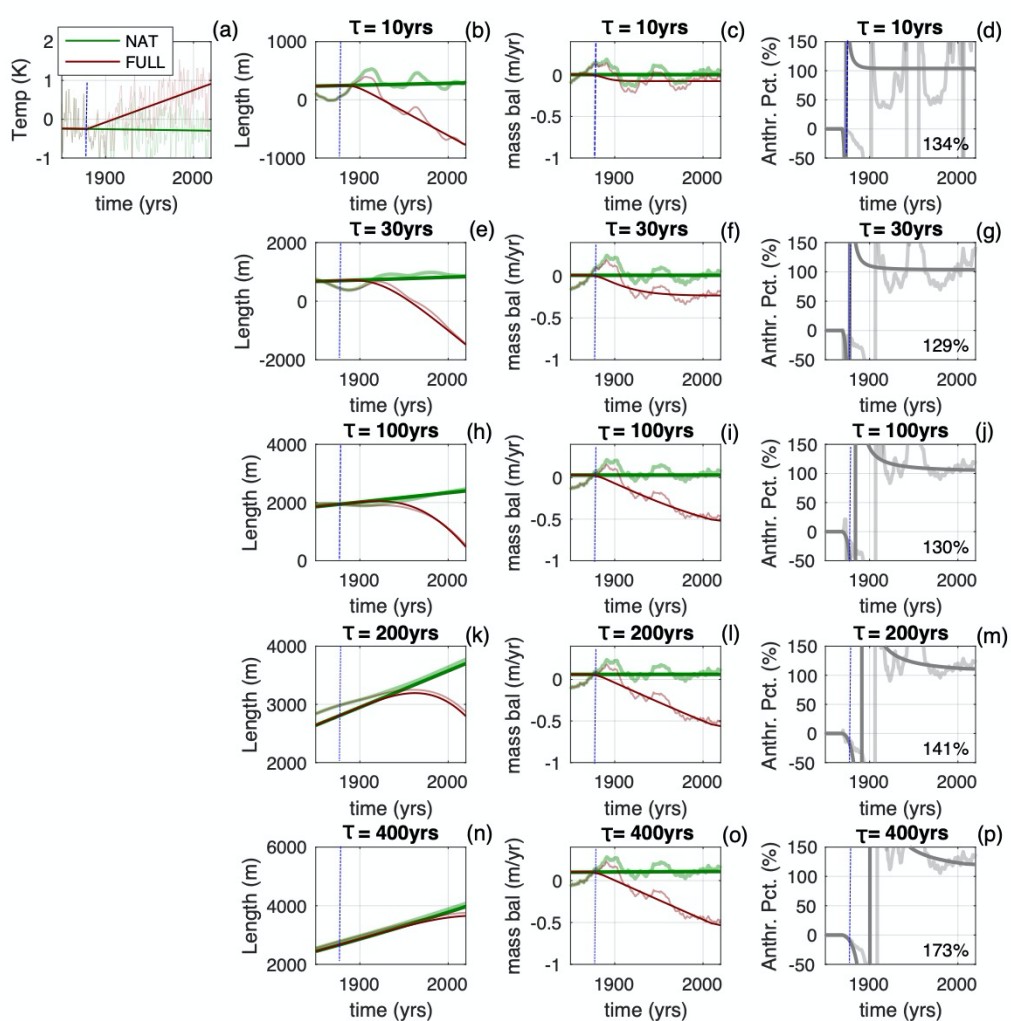

**Figure 3. Same as Fig. 2, but for the period 1850 to 2020. The percentages inserted into the right-hand columns give the anthropogenic cumulative mass loss relative to the total cumulative mass loss between 1880 and 2020. Note these cumulative percentages will not, in general, be equal to the time average of $F_{anth}$.**

The particular form of $F_{anth}$ depends on the assumed climate history. We consider three further synthetic LIA scenarios. In the first, the LIA cooling ends abruptly (i.e., temperatures jump back to their values at the start of the millennium) at the same time that the anthropogenic trend starts. This means that, in this scenario, the glacier would be shrinking during the 20th century even in the NAT case (Fig. 4, 5). In that case, the $F_{anth}$ curves still rapidly approach 100%, but from below rather than from above. For another, extreme, scenario, we assume an unrealistically deep LIA cooling that ends in 1750 (SM, Fig. S1, S2). The recovery from this cooling throws the glaciers into mass deficit that still exists in 1850 for the long-$\tau$ glaciers (similar to the deficit seen in the analysis of M14). But even in this extreme scenario (inconsistent with any observations), at the onset of the anthropogenic trend, $F_{anth}$ quickly rises towards 100%. Finally, we consider the possible impact of an unusually cold

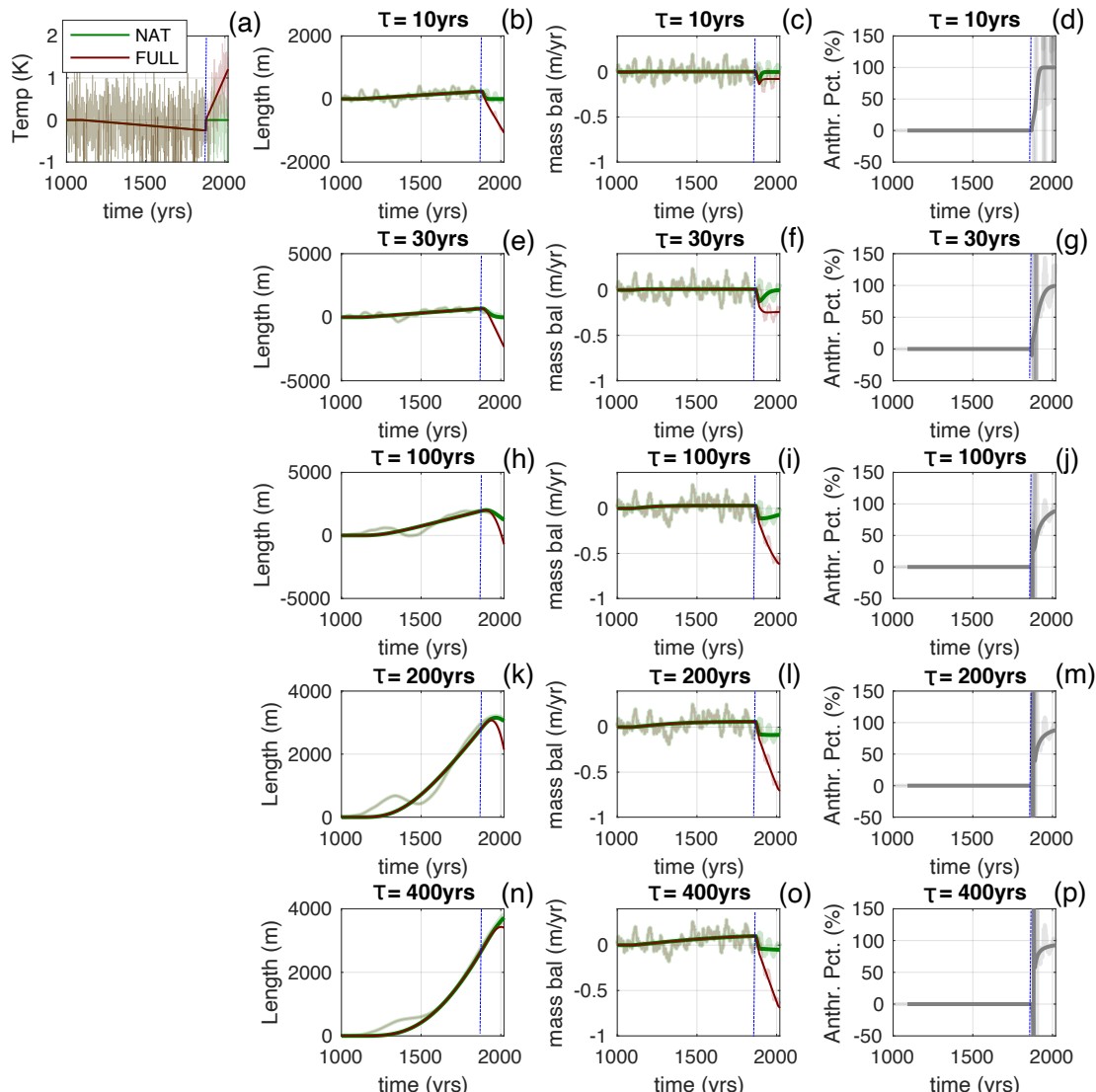

**Figure 4. Same as Fig. 2, but with an abrupt end to the millennial-scale cooling at the same as the onset of the anthropogenic trend.**

interval in the early 19[th] century. We add an intense top-hat-shaped cold event (0.75 °C) from 1800 to 1850, just before the onset of the anthropogenic trend (SM, Figs. S3, S4). This cooling is also deliberately much larger than is realistic. The cold event causes a spike of positive mass balance, and a noticeable advance of the short-$\tau$ glaciers (Figs. S3, S4). These glaciers are then already in a state of negative mass balance at the start of the anthropogenic warming. However, the impact of this initial condition is quickly forgotten and the magnitude of the cumulative anthropogenic mass loss is still $\geq\sim 80\%$ of cumulative total mass loss since 1850 for all $\tau$ considered.


Our examples have assumed a linear anthropogenic trend for simplicity, whereas it is thought that the attributable anthropogenic temperature change has increased more rapidly since 1970 (e.g., Haustein et al., 2019). We therefore calculated mass-balance histories for the industrial era using two piecewise-linear temperature trends. These showed similar $F_{anth}$ and cumulative anthropogenic mass loss to the results presented here (not shown).

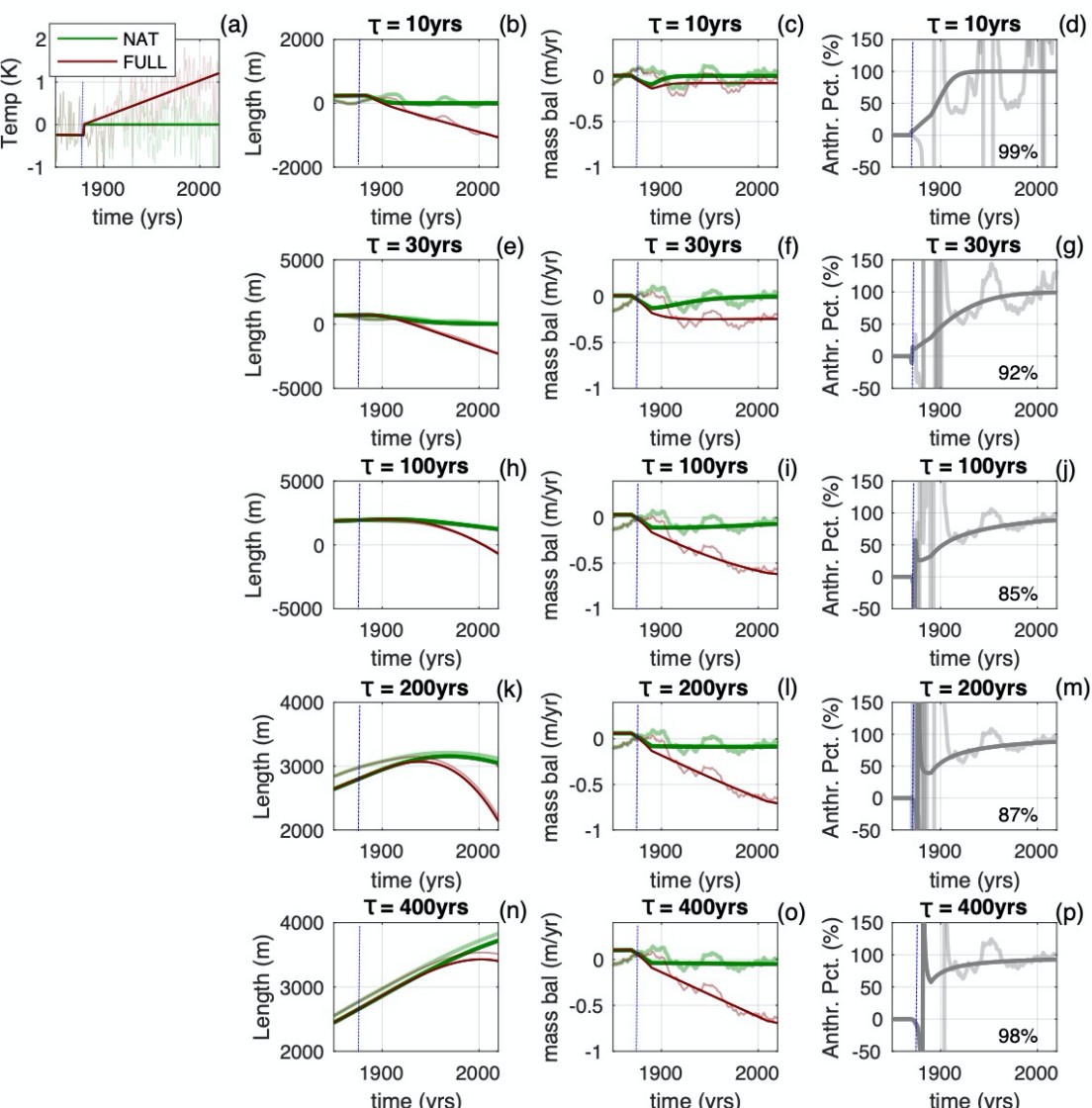

**Figure 5.** Same as Fig. 4, but zoomed into the period 1850 to 2020. The percentages inserted into the right-hand columns give the anthropogenic cumulative mass loss relative to the total cumulative mass loss between 1880 and 2020. Note these cumulative percentages will not, in general, be equal to the time average of $F_{anth}$.

The linear approximation for integrating total mass loss can affect the cumulative $F_{anth}$ because, if there is a natural component of mass loss early on, its impact on the cumulative mass loss depends on how much of the total glacier area is lost. We compared end-member cases where $L' \sim \bar{L}$, or $L' \ll \bar{L}$ after 150 years of forcing (SM, Figs. S5-S7). The linear approximation can introduce errors in mass loss if $L'$ approaches $\bar{L}$, but it only affects $F_{anth}$ if a large natural disequilibrium is assumed. Given that retreat since 1850 is much less than $\bar{L}$ for most glaciers, and especially large ones, assumptions about the climate history are much more consequential for evaluating $F_{anth}$.


Figure 6. (a) Tree-ring reconstruction of northern-hemisphere summertime temperature from Wilson et al., (2015). Also shown are observations of northern-hemisphere Apr-to-Sept temperatures from the GISTEMP v4 dataset. Other panels show the glacier length and mass balance as a function of glacier response time, $\tau$. Mass balance is shown with a 20-yr running mean applied.

These synthetic examples show that any temperature history that is even vaguely shaped like a hockey stick will have the same essential result for $F_{anth}$: during a period of rapid warming, the mass loss is overwhelmingly due to the glacier length lagging the warming climate; mass loss is only mildly affected by the inherited antecedent conditions from the prior slow cooling. We further demonstrate this point with more realistic temperature trajectories.

## 4.2 Proxy reconstructions of summertime temperature.

We next evaluate glacier response using proxy-based reconstructions of paleoclimate. Although this does not allow us to separate the natural and anthropogenic components of the mass balance, we can see the glacier response to the overall shape of the long-term cooling and subsequent anthropogenic warming. We use the recent tree-ring-based record of northern-hemisphere summertime temperature from Wilson et al. (2015), shown in Fig. 6a, which has the same overall shape as our synthetic millennium (Fig. 2a), with the addition of some low-frequency variability.

Observations from GISTEMP v4 dataset are shown for comparison (GISTEMP, 2020; Lenssen et al., 2019). The impact of different $\tau$s on the glacier length and on the mass-balance deficit over the last 150 years is clear, and the shapes of the length and mass-balance curves are similar to Fig. 2. If it is accepted that the rapid warming over the last 150 years is anthropogenic, then it can be anticipated that the implied $F_{anth}$ from Fig. 6 resembles the right-hand panels of Fig. 2.

**4.3 GCM ensembles of the last millennium**.

M14 used an ensemble of numerical climate simulations that disaggregated the natural and anthropogenic components of climate. However, the simulations were started at 1851, close to the start of the anthropogenic influence on climate. For glaciers, with their multi-decadal memory of prior climate history, the lack of model data from the preceding period leads to uncertainty about how to properly initialize those glaciers in 1851.


Here we make use of a recent millennial-scale ensemble of climate models that avoids the problem of preindustrial initialization. We use the Community Earth System Model (CESM) Last Millennium Ensemble (Otto-Bliesner et al., 2016). The FULL ensemble of simulations comprises thirteen model runs that span 850-2005 AD (Otto-Bliesner et al., 2016), and each simulation includes radiative-forcing contributions from volcanic aerosols, solar irradiance, orbital changes, greenhouse

gases, and anthropogenic ozone/aerosols, of the fifth-generation Coupled Model Intercomparison Project (CMIP5; Schmidt et al., 2011). There are also smaller ensembles for each individual forcing factor over the same period of time. Unfortunately, no integrations were made that included all the natural forcing factors but excluded anthropogenic forcing. That is to say, we do not have the direct equivalent of the NAT case of M14. However, of the natural forcing factors, volcanic aerosols have the largest impact in the CESM model (Otto Bliesner et al., 2016). So, to emulate the M14 approach, we consider the volcanic-

forcing ensemble ($\equiv$VOLC, four members) to be the equivalent of the NAT case in M14, and compare it to the ensemble mean of the FULL case.

We use the ensemble-mean, northern-hemisphere average, histories for summertime ($\equiv$ Apr. to Sept.) temperaure, $T$', and wintertime ($\equiv$ Oct. to Mar.) precipitation, $P$'. Following Roe and Baker (2016), in eq. (1) we set $b'_f = P' - \mu\phi T'$, where $\phi$ is

the ratio of the melt area to the total area of the glacier (for which we choose 0.8, although the exact value does not affect our conclusions, see also Section S5 in the SM). We apply the timeseries from these scenarios to calculate the glacier-length, mass-balance, and $F_{anth}$ (Figs. 7 and 8). With so few ensemble members, the ensemble mean still reflects a large degree of internal variability.

For volcanic forcing alone (VOLC), there is a slow cooling over the millennium as a result of the changing frequency of eruptions (Fig. 7a, Miller et al., 2012, Otto-Bliesner et al., 2016). When all forcings are included (FULL), there is also a ~0.6

°C warming starting around 1880, due to the anthopogenic $CO_2$ forcing. This is only about half what is seen in observations from GISSTEMP v4 dataset (Fig. 7a): the CESM is known to overestimate the cooling associated with anthropogenic aerosols

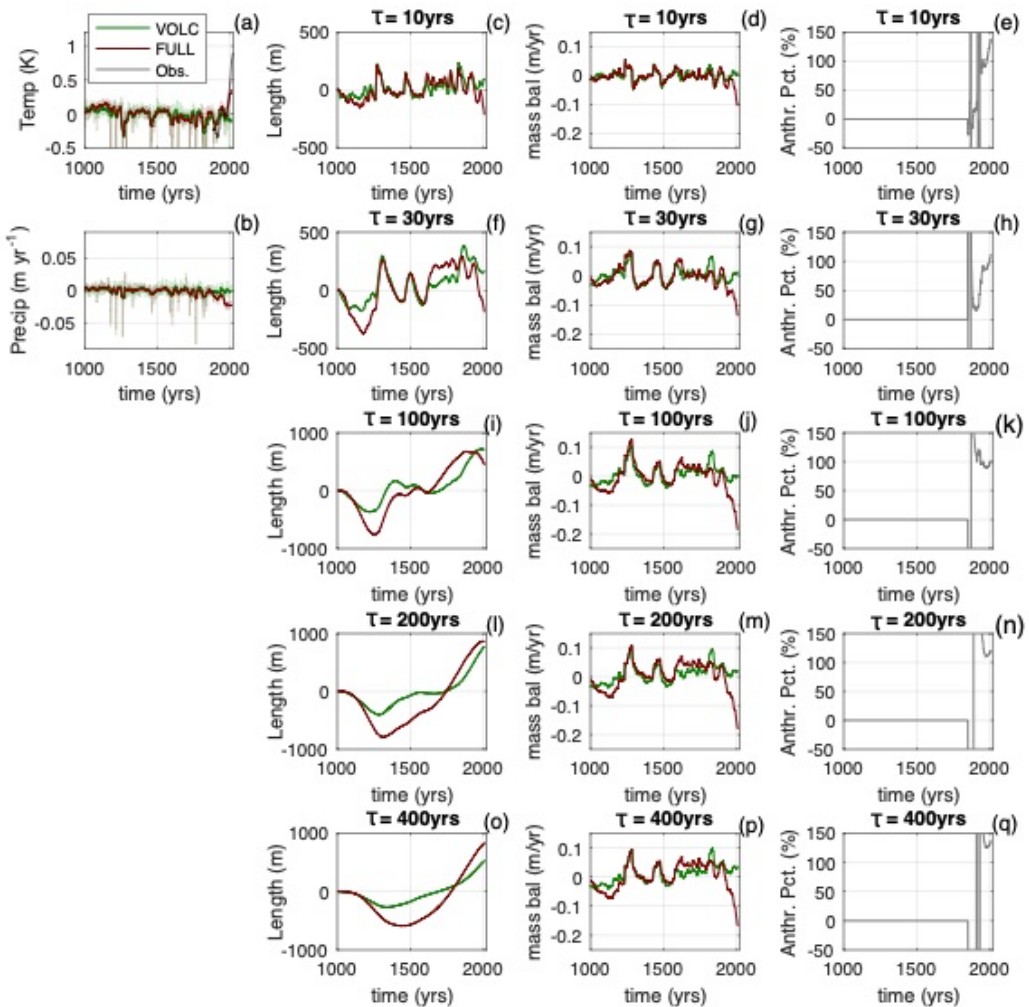

**Figure 7. As for Fig. 2, but from CESM millennium ensemble. (a) Northern hemisphere warm-season (Apr to Sept) temperature for the VOLC and FULL ensemble means, and compared to observations from GISTEMP v4 starting in 1880; (b) northern hemisphere cold-season (Oct to Mar) precipitation. Note the small variability and trend in the precipitation. The thicker lines in (a) and (b) are the 20-yr running mean. Others columns are the same as for Fig. 2, but here $F_{anth}$ is only calculated after 1850.**

(Otto-Bliesener et al., 2016). Neither the VOLC or FULL ensembles show strong precipiation trends or variability (Fig. 7b). Observed trends in average precipitation are generally weak and not yet statistically significant (e.g., Stocker et al., 2013). Moreover, century-scale observations and mass-balance models shows that temperature trends are more important than precipitation trends for driving trends in glacier mass balance (Roe et al., 2017, SM Sec[n]. S5).

Although the warming is considerably less than in observations, the fractional anthropogenic mass loss is similar to the

synthetic examples (Figs. 2, 3). The model simulations suggest that, in the absence of anthropogenic emissions, glaciers would have been stable or growing slightly (i.e., mass balance near zero, or slightly positive). As a result of this, $F_{anth}$, while somewhat noisy due to variability, on average exceeds 100% over the industrial era.

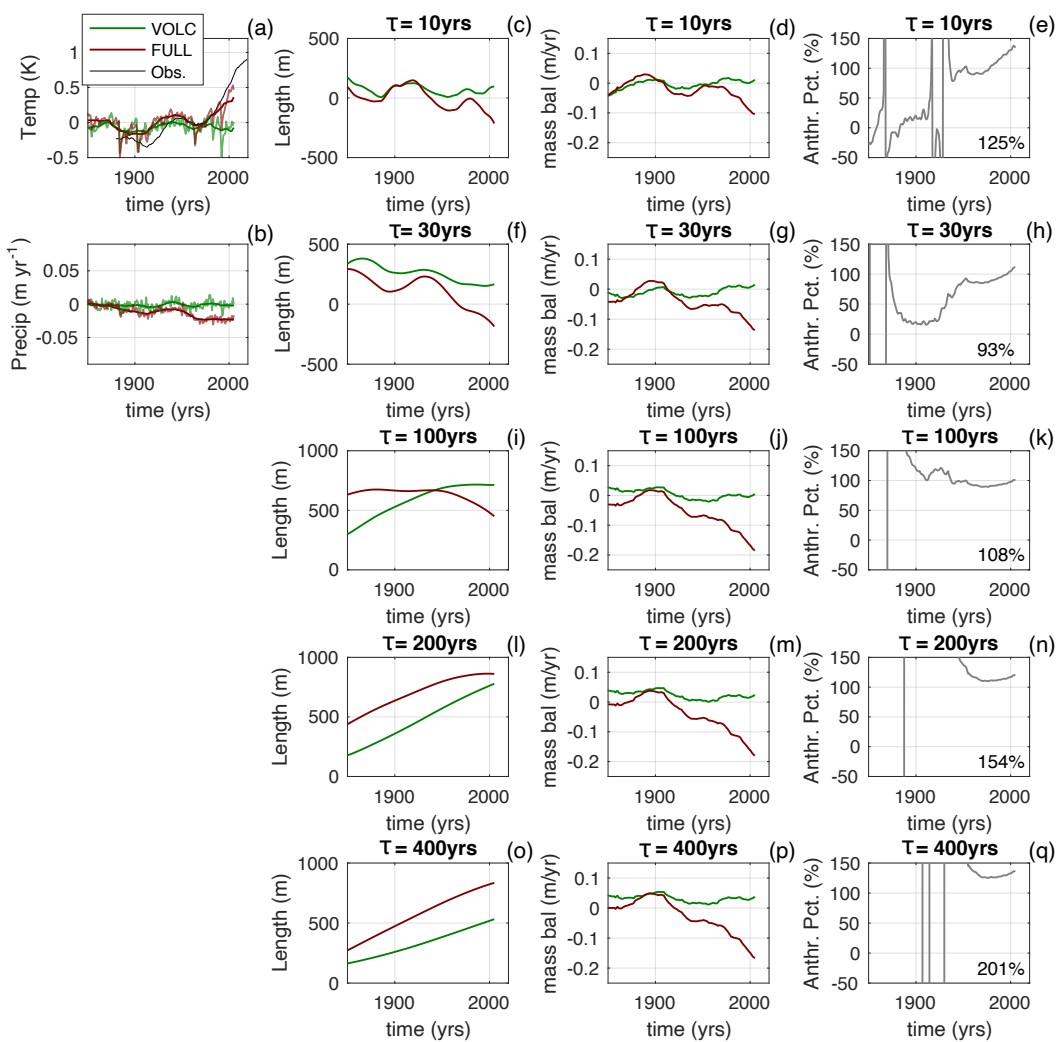

**Figure 8. Same as Fig. 7 but zoomed in to the 1850 to 2005 interval. The percentages inserted into the right-hand columns give the anthropogenic cumulative mass loss relative to the total cumulative mass loss between 1880 and 2020. Note these cumulative percentages will not, in general, be equal to the time average of $F_{anth}$.**

In Fig. S8 and S9, we present an example from a single grid point near the European Alps (47N, 10E). Precipitation variability

in particular is much larger at the local scale. Despite the larger local variability, $F_{anth}$ remains close to 100%, and it is

particularly clear for the large-$\tau$ glaciers. This is consistent with the result that, on centennial timescales, local temperatures are highly coherent with the global mean (e.g., Proistosescu et al., *in review*).

Finally, we can evaluate how much internal climate variability might affect estimates of the anthropogenic mass loss. Internal climate variability is represented by the differences among ensemble members, each of which has slightly different initial conditions in the year 850. This internal variability produces some differences in length and mass-balance histories (SM, Fig. S10), but in all cases, the mass balance of the FULL ensemble members emerges from the envelope of natural variability by the middle of the twentieth century. In the first decades after 1850, the ensemble of $F_{anth}$ curves is messy because the integrations without anthropogenic forcing have mass balance near zero, and individual members have singularities at different times. However, after about 1940 or so, the $F_{anth}$ curves consolidate to a narrower range of values, and the anthropogenic fraction of the total cumulative mass loss is narrowly clustered around the value obtained from the ensemble mean (i.e., Fig S10 cf. Fig. 8).

It is important to note that the CESM ensemble has some limitations for our purpose here. First, over the industrial era, the model produces too little warming relative to observations, perhaps because it overestimates the response to aerosols (Otto-Bliesner et al., 2016). Secondly, we approximated the counterfactual, all-natural climate using model integrations with volcanic forcing only; and with so few ensemble members the ensemble mean still reflects a lot of unforced internal variability. Lastly, of course it is just one specific climate model, and other models might yield different sensitivities to natural and anthropogenic forcing. Our results with CESM should therefore not be considered a definitive calculation, but rather as just one scenario. The analysis does highlight the importance of using milennial-scale integrations for charaterizing preindustrial glacier variability (e.g., Huston et al., in press), and for correctly initializing preindustrial glacier states: more millennial-scale ensemble integrations from different modeling groups would be valuable. Specifically, there is a need for numerical experiments with all natural forcings included simulataneously, and with large enough ensemble members to confidently diasaggregate the forced and unforced climate response.

## 4.4 Estimates of natural variability from instrumental records.

A final analysis we perform is to address the question of the attribution of cumulative mass loss from a slightly different angle. We estimate natural climate variability by detrending observations, and then ask how likely it is that the cumulative mass loss implied by observations might have occurred in a natural climate? In other words, we test the null hypothesis that the cumulative mass loss might have occurred due to natural variability alone. We illustrate this test for Hintereisferner in the European Alps (47N, 11E), following Herla et al. (2017). We take warm-season (Apr-to-Sept) temperature records from GISTEMP v4 for 1880 to 2019 (Fig. 9a), and annual-precipitation records from CRU TS4.04 (Harris et al., 2020), for 1900 to 2019 (Fig. 9b). Natural variability is estimated from the residuals after subtracting a third-order polynomial from the

temperature observations and a first-order polynomial from the precipitation. These residuals are then fit with a third-order
autoregressive process to capture the presence of any low-frequency persistence. Our results are not sensitive to the order of
the polynomial fit or the autoregressive process (from 1 to 5).

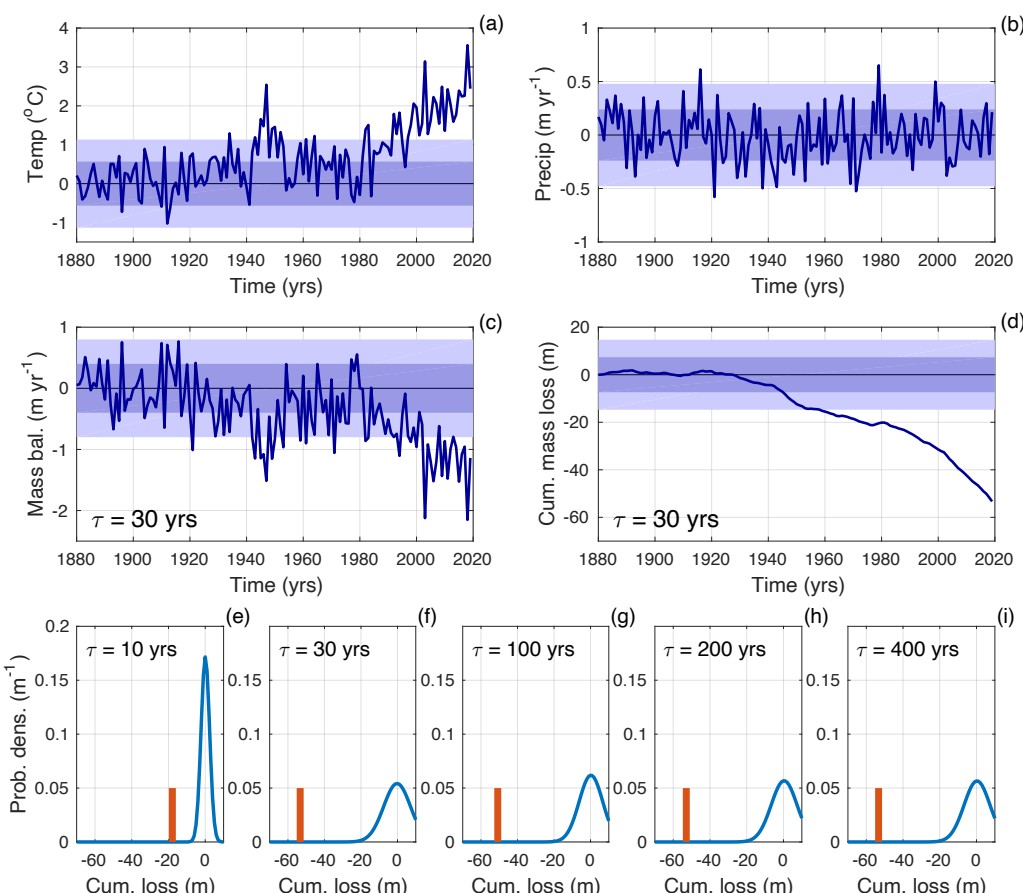

**Figure 9. Signal-to-noise ratios of cumulative mass loss, estimated from observations in the Alps (47N, 10E). (a) Summer (Apr-to-Sept) temperature anomalies from GISTMP v4 dataset; (b) Annual precipitation anomalies from the CRU TS4 dataset (values prior to 1900 were filled in with random values from the natural-variability statistics); (c) mass balance calculated using the 3-stage model with $\tau = 30$ yr; (d) cumulative mass loss using 3-stage model and $\tau = 30$ yrs; (e) to (i) the PDF of cumulative mass loss expected in any 140-yr period, based on estimated natural variability (blue line), compared to the cumulative mass loss calculated from the instrumental observations (orange bar). Each panel indicates the $\tau$ used in the 3-stage model. The shading in panels (a) to (d) indicates the 1 and $2\sigma$ bounds of the estimated natural variability, as described in the text.**

We next calculate the probability density function (PDF) of the cumulative mass change that might be expected in any random 140-yr period. We generate 10,000 synthetic 140-yr time series of precipitation and temperature that have the same statistical
properties as the estimated natural variability. We also vary the initial condition for the glacier (i.e., $L'$ at $t = 1880$). For this, we draw randomly from the PDF of the glacier's natural variability (Roe and Baker, 2014, 2016). For each 140-yr period, we

calculate the mass-balance (e.g., Fig. 9c) and cumulative mass loss (e.g., Fig. 9d) for five glaciers with each of our values of $\tau$. The statistics of what should be expected just from natural variability are compiled and then compared to the time series calculated from observations. The negative mass balance (Fig. 9c) due to the warming temperatures leads to an accumulation of mass loss that lies far outside what could have ocurred naturally (Fig. 9d). For each value of $\tau$, the chance of the cumulative mass loss arising from natural variability is much less than 1% (Figs. 9e to i). In IPCC language, with these assumptions and results, it is *virtually certain* that the calculated cumulative mass loss could not have happened without the climate changes attributed to anthropogenic emissions.

Alternatively, the natural variability could have been estimated from the unforced control runs from ensembles of GCMs, following, for example, Haustein et al. (2019) or Stuart-Smith et al. (in press), rather than from detrended observations, but it is clear that the conclusions would be the same. Although the summertime warming of the Alps is somewhat stronger than elsewhere (Hartmann et al., 2013), the basic character of the observations in Fig. 9a and 9b are similar to what is seen around the world. The PDFs in Figs. 9e to i resemble those in Roe et al. (2017) that compared observed glacier retreat to that expected from natural variability. The two measures are closely connected: glacier retreat plus the thinning of the still-extant ice has to add up to the cumulative mass loss.

What all our results point to is that, for these metrics of glacier change (i.e., retreat and mass balance), the dynamics of glacier response do not really affect the conclusions we can draw about attribution that come from analyzing the temperature alone: we already know that the temperature change over the industrial era is anthropogenic; temperature is an important control on glacier length and mass balance; temperature change is the primary driver of the industrial-era glacier changes we observe.

**5 Discussion**

It is now clear that anthropogenic emissions are the primary driver of industrial-era (~1850 to present) climate change. For the change in global-mean temperature since 1850, the central estimate of the anthropogenic warming is 100%, with uncertainty bounds of ±20% at the *likely* confidence level (Allen et al., 2018). Similar results have been obtained for temperature on regional and local scales (Haustein et al., 2019; Stuart-Smith et al., 2021*)*. Trends in average precipitation are generally small, and of secondary importance for trends in glacier mass balance; precipitation largely adds noise (Figs. 7, 8, SM Sec[n]. 5, Figs. S8, S9).

Essentially, over the industrial era, there are no known sources of natural climate forcing or internal variability that have a comparable impact to anthropogenic emissions. Local climate variability adds noise that can broaden the uncertainty in

attribution estimates. However, such local variability does not project very strongly onto 150-yr trends. Moreover, local effects tend to be cancelled out when global-mean changes are evaluated.

Like most of the modelling systems that downscale global climate information for glacier mass-balance and sea-level-rise predictions (e.g., Marzeion et al., 2020), we've assumed mass balance is a simple function of precipitation and temperature. Of course, in detail and in any given moment, glaciers are messy physical systems and the mass balance can be complicated. For instance, avalanches and wind-blown snow can be sources of mass input for small glaciers; and ablation is controlled by

455 the surface-energy balance, for which cloudiness, windiness, relative humidity, topographic shading, and valley-scale microclimates can all matter. These and other factors certainly play a role on sub-seasonal timescales; and for some settings, particularly arid tropical glaciers where sublimation is important (e.g., Mölg and Hardy, 2004; Rupper and Roe, 2008), care is needed to assess the mechanisms of change. But across most of the world and on centennial timescales, the links between increasing temperatures, rising equilibrium-line altitudes, and the consequent loss of ice, are straightforward and can be

confidently presumed.

We've focused here on the anthropogenic mass loss relative to the total mass loss from small ice caps and glaciers. This quantity depends only on two things: the anthropogenic contribution to the climate trend, and the glacier response time. We've explored a wide range of response times from 10 to 400 yrs, which covers the great majority of Earth's glaciers and small ice

caps. If, as per IPCC assessments, it is accepted that the best estimate of the magnitude of anthropogenic warming is 100% of the observed warming, then our results imply that the best estimate of anthropogenic mass loss is also 100% of the observed.

Our analyses have also shown that the relative contribution of the anthropogenic mass loss does depend to a degree on the inherited climate history. We made some extreme assumptions about climate history (SM, Figs. S1 to S4), and showed the

470 anthropogenic mass loss nonetheless still dominates the total. There is no objective algorithm for assessing uncertainty bounds on glacier loss: models are just models, and proxy reconstructions have attendant uncertainties, so uncertainty bounds must be based on assumptions. If we assess it is *likely* that, without anthropogenic forcing, climate history would have lain somewhere between a continued cooling (i.e., Fig. 3) and an abrupt end at 1850 (Fig. 5), then we would assess it is as *likely* that the cumulative anthropogenic mass loss lies between 85% and 180% of the total cumulative mass loss since 1880, and across all

475 of the glacier response times we've considered. Recall values can exceed 100% because glaciers might otherwise have been gaining mass. For the vast majority of alpine valley glaciers (that have $\tau \leq \sim100$ yrs), the *likely* range would be 85% to 130%. The contribution to sea-level rise must be weighted towards the larger, longer-$\tau$ glaciers and ice caps, for which the higher end of the uncertainty bounds might be considered. Note that these ranges are similar to that suggested by the variations among CESM ensemble members (SM, Fig. S10).

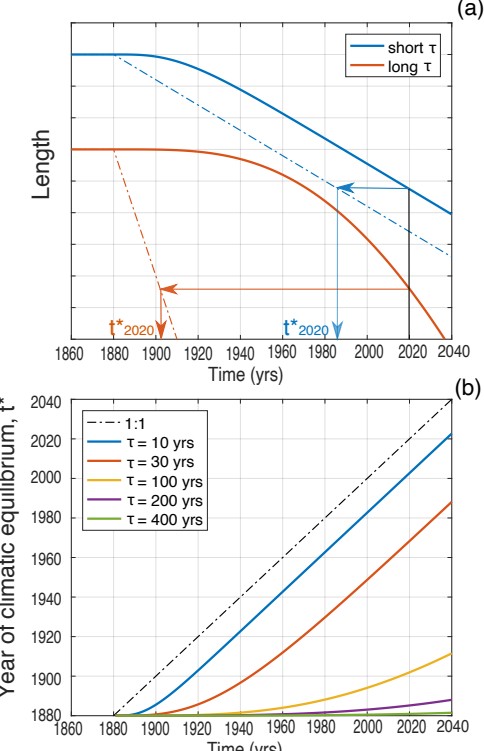

Why are these results so different from M14, with their fractional attribution of ~25% over the industrial era? M14 used similar GCMs to ours, so the anthropogenic contribution to climate change will be similar; and, as here, M14 used a mass-balance scheme based on temperature and precipitation. M14 analysed the full global inventory of glaciers in order to evaluate contributions to global sea-level rise. This goal required extrapolating from the current climate and glacier state, and some assumptions were necessary. Firstly, M14 assumed a response time based on the mass turnover within the glacier, $\tau \sim H/P$, rather than the Jóhannesson et al. (1989) geometric timescale, $\tau \sim -H/b_t$ (Eq. 2). Since, in many areas, average precipitation rates are lower than terminus ablation rates, M14 response times were larger than is probably correct. Secondly, M14 had to make an assumption about how the current glacier disequilibrium affects the initial states of their glaciers. It was assumed that there was a time in the past, $t^*$, at which the modern geometry would have zero mass balance given the climate at that time (i.e., the glacier would be in equilibrium). The value of $t^*$ was calibrated from the sparse network of glaciers with mass-balance observations, and those $t^*$s were extrapolated to all other glaciers, irrespective of their geometry. The assumption of mass-balance equilibrium at $t^*$ was then used to calibrate the individual ablation parameters for the unobserved glaciers (see M14, or Marzeion et al., 2012 for details). There is likely to be a bias in this procedure because $t^*$ is, in fact, a sensitive function of glacier response time: whereas small-$\tau$ glaciers are always in near-equilibrium because of

Figure 10. (a) Illustration of the concept of $t^*$ for a linear temperature trend starting in 1880. The thicker lines show the evolving glacier length as a function of time, for small-$\tau$ (blue) and large-$\tau$ (orange) glaciers. Dash-dot lines show the glacier-length that would be in equilibrium with the evolving climate as a function of time. For any given year, $t^*$ is found from the separation in time between these two lines; (b) Time of climatic equilibrium, $t^*$, for the length of a glacier subjected to a linear warming trend, shown as a function of time and response time $\tau$. Large-$\tau$ glaciers are far out of equilibrium from the modern climate (Christian et al. 2018). Curves calculated from Eq. 2.

their rapid adjustment, large-$\tau$ glaciers have barely begun to respond to the anthropogenic trend (Fig. 10a). For a linear trend, $t^*$ is equivalent to the lag behind the equilibrium response (Fig. 10a). In the limit of $t \gg \epsilon\tau$, $t^* \to t - 3\epsilon\tau$ (Christian et al., 2018). The variations of $t^*$ with $\tau$ and time are shown in Fig. 10b. Even in 2020, the geometries of very large-$\tau$ glaciers are in near-equilibrium with the climate of the preindustrial age. They have a lot of catching up to do, even to adjust to the current climate.

Any effort to downscale climate predictions to all glaciers globally must deal with the problem of initialization. In many cases $\tau$ is quite uncertain because thickness is poorly constrained, and long mass-balance records are available only for a handful of


glaciers. Any downscaling algorithm must also correctly represent the glacier's dynamic response function: for instance, the difference between assuming $\tau$ represents an exponential adjustment instead of a three-stage (or sigmoidal) adjustment can substantially affect the assessed degree of disequilibrium (Christian et al., 2018; Fig. 10). Assumptions to deal with initialization still vary widely among down-scaling studies (Marzeion et al., 2020).

M14 suggests there would have been a sustained mass-balance deficit over the past 150 years, even in the absence of anthropogenic forcing (i.e., in the NAT case). The comparatively small anthropogenic contribution to total mass loss in M14 arises because it is not until 1990 that the anthropogenic contribution overtakes this natural mass-balance deficit. However, as demonstrated here, a sustained mass-balance deficit over the past 150 years requires an ongoing warming trend, which is not seen in the naturally forced model simulations (e.g., green lines); or it requires a dramatic antecedent disequilibrium that 525 continues to affect the larger $\tau$ glaciers. Moreover, the degree of disequilibrium would have to be larger than even the unrealistic LIA scenario in Fig. S1, and that is not consistent with known climate histories (e.g., Fig. 6). Here, we've circumvented the need to estimate the initial condition of pre-industrial glaciers by considering the response over the whole millennium. Furthermore, by not attempting to simulate the full global inventory of glaciers, we can put more focus on the specific question of the attribution to anthropogenic climate forcing. We can also distinguish and emphasize the different 530 behaviours of short-$\tau$ and long-$\tau$ glaciers, rather than drawing blanket conclusions from the global aggregate. Our results cast the anthropogenic mass loss in terms relative to the observed mass loss. As such, we do not assess the absolute magnitude of mass loss from small ice caps and glaciers, but our results should affect the interpretation of what the cause of that mass loss has been.

The tendency of a glacier to adjust towards a mass-balance equilibrium with a characteristic response time is a robust piece of physics. Our results depend only on that physics, and on our knowledge of the magnitudes and rates of climate change over the past millennium. We conclude that the research community should feel confident in making much stronger statements about the role of anthropogenic climate change in the loss of glacier mass. Following the most-recent IPCC statements on temperature changes since 1850 (Allen et al., 2018), our calculations suggest that the central estimate of the anthropogenic 540 mass loss of glaciers and small ice caps is essentially 100% of the observed. From our results, we assess a *likely* uncertainty range of 85% to 130% for Alpine valley glaciers, and 85% to 180% for larger ice caps with multi-century response times. Consistent with assessments about the cause of industrial-era temperature change (e.g., Allen et al., 2018; Haustein et al., 2019), it is extremely likely that anthropogenic emissions have been the primary driver of the cumulative mass loss, and hence the contribution to sea-level rise, since 1850. Our conclusions are consistent with previous research showing that observed 545 glacier retreat is categorical evidence of industrial-era climate change (Roe et al., 2017). Finally, our results fully justify using the iconic imagery of centennial-scale glacier retreat around the world as a powerful visualization of just how far these icy landscapes have been altered by human activity.

**6. Supplement Link**. To be added


**7. Author contributions.** G.H.R., J.E.C, and B.M. designed the study, performed the analyses, and wrote the manuscript.

**8. Competing interests.** The authors declare that they have no conflicts of interest.

**9. Acknowledgements.** We thank Alan Huston for help with the CESM data; and Marcia Baker, Kyle Armour, and Rupert Stuart-Smith for valuable suggestions and comments.

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
