# Peer review of "On the attribution of industrial-era glacier mass loss to anthropogenic climate change"

_The Cryosphere, 2020_

## Referee Comment (RC1) · Lauren Vargo (Referee) · 2 Dec 2020

This manuscript investigates the influence of anthropogenic forcing on glacier mass loss over the industrial era. The authors use a 3-stage model to simulate glacier length and mass balance changes for glaciers with different response times. I'm less familiar with this model compared with other glacier models, but it was well-described and the original paper was cited to better understand the model. The authors use different climate inputs: synthetic temperature, proxy temperature, and model temperature and precipitation. These climate inputs cover the last millennium to ensure that glaciers, even those with response times of 400 years, have reached climatic equilibrium over the industrial period for which anthropogenic forcing is calculated. Running these longer simulations is where this paper builds on previous attribution studies, particularly Marzeion et al., 2014.

This paper is well-written, the structure makes it easy to read, the figures are clear and add to the manuscript, and the paper provides insight into an important topic: investigating the influence of anthropogenic forcing on glacier mass loss, in a way that includes inherited conditions.

The main question/concern I have is if the anthropogenic contributions are calculated in an appropriate way. I understand, mathematically, why presented anthropogenic forcing values can exceed 100% (if a glacier losing mass in full forcing scenario would have gained mass in natural forcing). But, if you're looking to calculate the anthropogenic contribution to glacier mass loss, should you do so in a way that calculates that contribution to be between 0 and 100%? As negative values and values over 100% don't realistically make sense. I think calculating the influence of anthropogenic forcing in another way so that values are between 0 and 100% would make the paper and results clearer and less confusing, giving it a higher impact. I wonder if there is a better way to do this using cumulative mass change from ~1880 through the end of the simulation. It also seems important that a small change in temperature between that shown in Figures 2 & 3 to Figures 4 & 5 do impact the contribution of anthropogenic forcing percentages by 1/3 to almost 1/2 for different response times.

Specific comments (intermediate):

L49-50: This line makes it feel like this is an important part of this paper- assessing all glaciers for the contribution to sea level rise. Should this go in the title, or at least abstract? But then I was waiting for the results to be tied in with sea level rise later on, but it wasn't discussed.

Overall, I wonder if it's important to emphasize that these are idealized scenarios: all glaciers globally are represented by just five different response times, and using a model that doesn't include
L308-316: Are the solar irradiance and orbital changes ensembles not analyzed at all? If you only use one and volcanic-forcing has the biggest impact, using that one makes sense. But why not include all (besides too much data/run time)?

L419: This goes along with my main comment, but results show anthropogenic contribution of over 100%.

Specific comments (minor/technical):

L35-36: Do we know this for sure/are there studies that have shown this? Is there a citation to use?

L41: "An alternative approach", referring to an alternative to Roe et al., 2017?

L46: Vargo et al 2020 shows a method that applies the method discussed here but to glaciers with limited observations (but still needs some observations, so still a subset of the total).

L58: Anthropogenic forcing would account for some observed retreat? 25% of mass loss due to anthropogenic forcing isn't nothing.

L61-70: Good setup/description for part of the problem you're addressing

L159-174: I think these assumptions are reasonable (some need to be made). But for Arctic glaciers, if 10% have a response time of over 400 years, those are probably some with the most ice mass? With larger contributions to sea level rise than smaller glaciers with quicker response times? Maybe just something to note.

L197: I wonder if/how much changing the 0.5C of white noise influences results? Maybe discussed in Roe et al., 2017.

L245: This paragraph is a nice overview.

L249: 'zThe' typo

Section 4.2: I know papers have shown that for many glaciers around the world, summer temperatures are especially important in overall mass balance. But was a summer temperature reconstruction used because that's what is available, or because it's the best value to use?

L319: Why is precipitation considered now but not with idealized scenarios? And how is it incorporated here versus in equations for the idealized scenario?

L360-364: I haven't seen this done before, but I think it makes sense.

L378 - 379: Seems odd to say it could have been done a completely different way (different input data) but results would obviously be the same. Is it obvious?

Fig 9: Is it useful to compare where measured mass loss falls/plots?

Figure 10: This took me reading a couple of times to understand, but it helps show that a problem with M14 is that glaciers with long response times (100, 200, 400 years) haven't actually reached climatic equilibrium.

---

## Referee Comment (RC2) · Jonathan Bamber (Referee) · 9 Dec 2020

Review of Roe, Christian and Marzeion 2020

**General comments**

This is an important topic of very high relevance to the journal and science community. The paper is clear and well written. The central message is clear but the way it is presented is, in my view, misleading and unhelpful as explained below and this is my only major issue with the paper.

The issue relates to the definition of the proportion (fraction) of mass **change** (not loss) that is due to anthropogenic climate change (ACC) versus natural which will I've called internal variability (IV). It is unhelpful to define the fractional change in the way the authors have done. As they point out this leads to infinite fractional values for ACC. If the authors plotted the IV fraction this could also have an infinite value when FULL passes through zero, clearly nonsensical and unhelpful in providing clarity in the central message of the paper. When FULL = 0, it means that IV=ACC but with opposite signs. It would seem reasonable to define that as IV and ACC having an equal, fractional contribution to the instantaneous mass balance (mb), but the definition used does not do this and leads, therefore, in the subsequent plots of cumulative mb attribution to misleading values. I do not dispute the fundamental message that almost all of the industrial era mass loss from glaciers is due to ACC (see final comment in this section) but the way this has been defined and presented, does not help.

Attribution of temperature change is relatively challenging and there have been several different approaches used, which result in somewhat different contributions from ACC and IV to temperature anomalies since 1880. This matters, because the proportional contribution from ACC and IV will directly impact the proportion assigned to glacier mass balance. Below is one example of attribution from (Sévellec and Drijfhout 2018). The linear ACC trend from 1880-1920 is about half the IV trend for the same period. Using a different definition for fractional contribution to mb would result in ~66% of the cumulative instantaneous anomaly being due to IV and ~33% to ACC (approximately). Let's assume that from 1920-2005, the contribution from ACC is 100% and IV 0% (i.e. the sum is 1), which is roughly supported by the figure below. Over the full time period this gives a 22% contribution from IV and 78% from ACC. This is assuming an unrealistic instantaneous response and is given simply to illustrate that the way the authors have chosen to define the fractional attribution is, in my view, unhelpful.

Fig 8c suggests that the cumulative mb anomaly from 1850-1920 is ~ zero, for the case where τ=10 yrs, while the ACC % is 125% for the full period from 1850-2005, which is, again, counter intuitive. It also doesn't seem to square with the integral of Fig 8e if the extreme values are ignored. In this case, the ACC % only exceeds (in a meaningful way) 100% from ~1960 onward. I do not understand, therefore, how a value of 125% is achieved, or in what way It is a meaningful way of representing the ACC contribution to mass change for this example. Likewise for the integral of 8h. For these shorter time constant glaciers, the ACC % only approaches 100% in the second half of the 20th C. This is not so far from the inference made in M14, at least for the short τ glaciers.

For the longer time constant glaciers, presumably there is a +ve mb memory locked in that compensates for the >100% ACC percentage but this somehow contradicts the authors' own claims in the abstract: "the anthropogenic component of the mass loss is essentially 100%." In the case of large τ, the authors are stating it is 200%. If the authors stand by the values in Fig 8 then the ACC contribution for all glaciers is significantly >100% but nowhere is that stated or claimed for reasons that I believe the authors know themselves.

In many respects, I found Fig 9 a more informative and clear demonstration of the role of ACC in post industrial glacier mb trends alongside the sentence starting at line 385.

**Fig. 1**

[Figure]

Attribution of observed global-mean surface air temperature (GMT) and sea surface temperature (SST). **a, b** The total (red) annual, (purple) 5-year and (blue) 10-year variations in GMT and SST measured from 1880 are decomposed (through an attribution method based on multivariate linear regression onto volcanic eruptions, aerosol concentration, and greenhouse gas concentration[2]) into **c, d** a forced contribution and **e, f** a residual. **g, h** Relative variance of forced and residual GMT and SST changes as a function of the duration of these changes. Variations are mainly controlled by the residual, rather than forcing on interannual to decadal timescales. The observed GMT are from NASA GISS temperature data, and SST is from the NOAA ERSSTv5 record

**Technical comments**

In the synthetic temperature figures (e.g. Fig 1) it wold be helpful to include a vertical dashed line in the length and mb columns indicating the start of the perturbation in temperature.

Eqn 1. Replace = with ≈.

L299-300. This statement is misleading. The synthetic temperature experiments are useful to illustrate a point but they are **not** representative of the true temperature anomalies over the last 150 years and, in particular, the gradient of the ACC temperature trend over that time period. See Figs 1a and c above. There is a change in gradient from ~1960 onward. Something that is sort of apparent in Fig 9a and sort of implicitly captured in Figs 8e and h but not discussed at all.

L333 "The models" => The model.

L373 Fig 7e to I => Fig 9 e to i

L418 See previous comment. The definition used means it is > 100%, and for long $\tau$ glaciers closer to 200%.

There is a further inconsistency in the way this study has undertaken the difficult attribution part of the problem. As far as I can tell, the decline in length and mb in Fig 6 begins in ~1820, which is consistent with some temperature reconstructions for the last 2 millennia, which have an increase starting ~1800 of ~0.2 degs. That pre-dates any ACC contribution unlike the expt shown in Fig 5 and is not apparent in any other plots because they all start in 1850.

**References**

Sévellec, F. and S. S. Drijfhout (2018). "A novel probabilistic forecast system predicting anomalously warm 2018-2022 reinforcing the long-term global warming trend." Nature Communications **9**(1): 3024.

---

## Author Comment (AC1) · 2 Jan 2021

Review 2:
We thank Dr. Lauren Vargo very much for her time in reviewing our manuscript, and for her comments and suggestions. We have responded with comments interspersed into her review.

This manuscript investigates the influence of anthropogenic forcing on glacier mass loss over the industrial era. The authors use a 3-stage model to simulate glacier length and mass balance changes for glaciers with different response times. I'm less familiar with this model compared with other glacier models, but it was well-described and the original paper was cited to better understand the model. The authors use different climate inputs: synthetic temperature, proxy temperature, and model temperature and precipitation. These climate inputs cover the last millennium to ensure that glaciers, even those with response times of 400 years, have reached climatic equilibrium over the industrial period for which anthropogenic forcing is calculated. Running these longer simulations is where this paper builds on previous attribution studies, particularly Marzeion et al., 2014.

This paper is well-written, the structure makes it easy to read, the figures are clear and add to the manuscript, and the paper provides insight into an important topic: investigating the influence of anthropogenic forcing on glacier mass loss, in a way that includes inherited conditions.

The main question/concern I have is if the anthropogenic contributions are calculated in an appropriate way. I understand, mathematically, why presented anthropogenic forcing values can exceed 100% (if a glacier losing mass in full forcing scenario would have gained mass in natural forcing). But, if you're looking to calculate the anthropogenic contribution to glacier mass loss, should you do so in a way that calculates that contribution to be between 0 and 100%? As negative values and values over 100% don't realistically make sense. I think calculating the influence of anthropogenic forcing in another way so that values are between 0 and 100% would make the paper and results clearer and less confusing, giving it a higher impact. I wonder if there is a better way to do this using cumulative mass change from _1880 through the end of the simulation. It also seems important that a small change in temperature between that shown in Figures 2 & 3 to Figures 4 & 5 do impact the contribution of anthropogenic forcing percentages by 1/3 to almost 1/2 for different response times.

Both reviewers raised the issue of this fractional metric. We've specifically chosen it because it is used extensively by the IPCC; and it was used by Marzeion et al (2014, M14), whose analysis we parallel. Moreover, the headline result of M14 was given in terms of this metric, and the result has been cited in subsequent IPCC reports and elsewhere in the literature. The word 'contribution' causes a strong reaction in some, especially when the number is negative or exceeds 100%, even if the equation hasn't changed. We note that the original M14 analysis allowed for a negative value in the uncertainty range and that, for anthropogenic warming, the current IPCC uncertainty range allows for it to exceed 100% of the observed. Apparently, there were divergent opinions in IPCC discussions over the use of word "contribution" and reports now avoid the word; and cast the assessment as the magnitude of anthropogenic warming relative to the observed. In our revisions we adopt this: we have removed the word contribution from the manuscript anywhere it might have this connotation. And we describe the analysis everywhere as the magnitude of the anthropogenic mass loss relative to the observed (or equivalently, relative to the total). In addition to these changes throughout the manuscript, when we introduce the metric, we now give readers two substantial paragraphs detailing the reasons for the metric, guiding them about how to interpret it, and now tell them ahead of time that we also analyze another metric in a later section. We hope that this sets a reader up better to interpret our results.

The two paragraphs are reproduced here: *"We here take a moment to discuss various possible metrics that can be used to characterize the influence of anthropogenic factors on glacier mass balance. Each have different merits. One approach is to set up a null hypothesis of a counterfactual, purely natural climate with no anthropogenic forcing, and to pose the question "how likely is the observed change in the natural case?". If the likelihood of the observed change falls below some stated level of significance, then the null hypothesis is rejected, and a "signal" can be declared detected. This was the approach applied to glacial retreat in Roe et al., 2017, and we apply it to mass balance in Section 4.4 of this paper. Another approach is to define a measure of the variance explained among an ensemble of climate models where natural variability, different anthropogenic forcing scenarios, and different models all contribute to the spread (e.g., Hawkins and Sutton, 2009; Marzeion et al., 2020), which is helpful for partitioning sources of uncertainty. A third approach, one that has been used by the Intergovernmental Panel on Climate Change (IPCC) over several past reports, is to estimate the magnitude of the anthropogenic component relative to the observed change. For instance, as of the 2018 IPCC Special Report, the central estimate of the magnitude of the anthropogenic change in global-mean surface temperature over the industrial era is that it is equal to 100% of the observed warming, with an assessed likely (i.e., a 2-in-3 chance) range of $\pm20\%$ (Allen et al., 2018). Note this uncertainty range allows for the possibility that the magnitude of the anthropogenic warming exceeds 100% of the observed, because it is possible that the natural climate would have otherwise been cooling. M14 followed the IPCC in using this approach. It is useful, for instance, in the context of interpreting sea-level rise, where other contributions from thermal expansion and large ice sheets need to be understood. We want our results here to parallel those of M14, and we adopt the same approach as M14 in Sections 4.1 and 4.3 of this study.*

*M14 defined the following metric for comparing the magnitude of the anthropogenic mass loss to the FULL mass balance: $F_{anth} = 100\% \times (ANTH/FULL) = 100\% \times (1 - NAT/FULL)$. M14 calculated the 20-yr running mean of $F_{anth}$, in order to assess how anthropogenic mass loss has evolved over time at decadal scales. For assessing the total change in mass over the industrial era, ANTH and FULL can be integrated over time before calculating their ratio. Note that this cumulative value will not, in general, be equal to the time-average of $F_{anth}$. In our analyses, we present both the decadal and cumulative values. Both provide useful information: the 20-yr running mean provides insight into the causes of decadal-scale mass-balance observations, whereas the cumulative anthropogenic mass loss can be more directly tied to glacier retreat (see Section 4.4) and sea-level rise."*

Specific comments (intermediate):

*L49-50: This line makes it feel like this is an important part of this paper- assessing all glaciers for the contribution to sea level rise. Should this go in the title, or at least abstract? But then I was waiting for the results to be tied in with sea level rise later on, but it wasn't discussed.*
We specifically don't tie into sea-level rise with our main results: we can make strong statements about the relative importance of natural variability and anthropogenic climate change in mass loss, but we do not estimate absolute mass loss. However, our results should affect interpretations of what has caused the sea-level rise associated with mass loss from ice caps and glaciers, and we now include a specific sentence about this in the discussion to clarify.

*Overall, I wonder if it's important to emphasize that these are idealized scenarios: all glaciers globally are represented by just five different response times, and using a model that doesn't include*
The end of this comment got clipped out of the review, unfortunately. In one limit, all models are idealized scenarios. Our central result is showing that the magnitude of the anthropogenic mass dominates the total mass loss for all glacier response times. That is arguably a more powerful result than highly detailed simulations of a specific setting: it is more general, and shows that those very specific details (which have their own uncertainties of course), don't actually matter to understand the overall role of anthropogenic climate change in glacier mass loss.

*L308-316: Are the solar irradiance and orbital changes ensembles not analyzed at all? If you only use one and volcanic-forcing has the biggest impact, using that one makes*

*sense. But why not include all (besides too much data/run time)?*
There are not many millennium-scale GCM ensembles (more are gradually becoming available). The group that made these integrations did consider irradiance and orbital changes, but only individually, and then all together when they also included the anthropogenic $CO_2$. That is, there is not a run with all natural forcings and no anthropogenic forcing. It also is a GCM that warms less than observations over the 20[th] century. That is less than ideal for us, but doesn't prevent the essential point being made. We've included an expanded discussion of the model to guide a reader more helpfully.

*L419: This goes along with my main comment, but results show anthropogenic contribution of over 100%.*
Hopefully our reframing of the language around 'contribution' helps in the revised manuscript.

**Specific comments (minor/technical):**

*L35-36: Do we know this for sure/are there studies that have shown this? Is there a citation to use?*
We tweaked the language to make it clear it comes from Zemp et al. (2015).

*L41: "An alternative approach", referring to an alternative to Roe et al., 2017?*
Thank you, we've tweaked the language to clarify.

*L46: Vargo et al 2020 shows a method that applies the method discussed here but to glaciers with limited observations (but still needs some observations, so still a subset of the total).*
Thanks. Included as a reference.

*L58: Anthropogenic forcing would account for some observed retreat? 25% of mass loss due to anthropogenic forcing isn't nothing.*
Agreed, but if that was all it was (25%), it would stand in stark contrast to the use of glacier imagery to educate about the impact of anthropogenic climate change. Obviously, we hope our study resolves that apparent contradiction.

*L61-70: Good setup/description for part of the problem you're addressing*
Thank you!

*L159-174: I think these assumptions are reasonable (some need to be made). But for Arctic glaciers, if 10% have a response time of over 400 years, those are probably some with the most ice mass? With larger contributions to sea level rise than smaller glaciers with quicker response times? Maybe just something to note.*
Thanks, yes, we note in the discussion that for sea-level change, the upper-bound of the uncertainty estimates might need to be considered. Our central result doesn't change because a 400-yr timescale is already effectively at the large-tau limit.

*L197: I wonder if/how much changing the 0.5C of white noise influences results? Maybe discussed in Roe et al., 2017.*
Our equation is linear, so the amplitude of the noise leaves the impact of the underlying trend untouched. In other work (including in Roe et al., 2017), we have considered the impact of persistence in interannual variability, which can drive larger natural variability of the glaciers. Observed interannual variability in summer temperature and winter precipitation is quite close to white noise (see, e.g., Burke and Roe, Climate Dynamics, 2013)

*L245: This paragraph is a nice overview.*
Thank you!

*L249: 'zThe' typo*
Thank you!

*Section 4.2: I know papers have shown that for many glaciers around the world, summer temperatures are especially important in overall mass balance. But was a summer temperature reconstruction used because that's what is available, or because it's the best value to use?*
We chose it in preference to annual-mean reconstructions because it was a relatively recent publication, and because of summer's importance for melt. For the purpose here, we could have used several others equally well.

*L319: Why is precipitation considered now but not with idealized scenarios? And how is it incorporated here versus in equations for the idealized scenario?*
We omit precipitation for the idealized scenarios because precipitation mainly acts as a noise maker, rather than affecting trends. We included a section in the supplementary explaining why temperature trends dominate mass-balance trends. We now point readers to this section, and discussion why we omit precipitation when introducing the synthetic scenarios. For these experiments based on climate model output, we included precipitation for completeness and we now describe in the main text how precipitation is included.

*L360-364: I haven't seen this done before, but I think it makes sense.*

*L378 - 379: Seems odd to say it could have been done a completely different way (different input data) but results would obviously be the same. Is it obvious?*
The magnitude of interannual variability is sufficiently well represented by models, which is all the results depend on. We changed to use the word "conclusion" rather than answer

*Fig 9: Is it useful to compare where measured mass loss falls/plots?*
It is a nice idea. However, we haven't tried to simulate any specific glacier. In principle we certainly could, but then we'd really just be calibrating parameters to get agreement. We don't have direct mass-balance measurements going back to 1880, although reconstructions from 1865 for four glaciers in the Alps give similar magnitudes of cumulative mass balance (see Huss et al., 2008, JGR, Fig. 6).

*Figure 10: This took me reading a couple of times to understand, but it helps show that a problem with M14 is that glaciers with long response times (100, 200, 400 years) haven't actually reached climatic equilibrium.*

---

## Author Comment (AC2) · 2 Jan 2021

We thank Dr. Jonathan Bamber very much for his time in reviewing our manuscript, and for his comments and suggestions. We have responded with comments interspersed into his review. Review is in green, response is in black.

Review of Roe, Christian, and Marzeion, 2020
**General comments**
This is an important topic of very high relevance to the journal and science community. The paper is clear and well written. The central message is clear but the way it is presented is, in my view, misleading and unhelpful as explained below and this is my only major issue with the paper.
The issue relates to the definition of the proportion (fraction) of mass **change** (not loss) that is due to anthropogenic climate change (ACC) versus natural which will I've called internal variability (IV). It is unhelpful to define the fractional change in the way the authors have done. As they point out this leads to infinite fractional values for ACC. If the authors plotted the IV fraction this could also have an infinite value when FULL passes through zero, clearly nonsensical and unhelpful in providing clarity in the central message of the paper. When FULL = 0, it means that IV=ACC but with opposite signs. It would seem reasonable to define that as IV and ACC having an equal, fractional contribution to the instantaneous mass balance (mb), but the definition used does not do this and leads, therefore, in the subsequent plots of cumulative mb attribution to misleading values. I do not dispute the fundamental message that almost all of the industrial era mass loss from glaciers is due to ACC (see final comment in this section) but the way this has been defined and presented, does not help. Attribution of temperature change is relatively challenging and there have been several different approaches used, which result in somewhat different contributions from ACC and IV to temperature anomalies since 1880. This matters, because the proportional contribution from ACC and IV will directly impact the proportion assigned to glacier mass balance. Below is one example of attribution from (Sévellec and Drijfhout 2018). The linear ACC trend from 1880-1920 is about half the IV trend for the same period. Using a different definition for fractional contribution to mb would result in ~66% of the cumulative instantaneous anomaly being due to IV and ~33% to ACC (approximately). Let's assume that from 1920-2005, the contribution from ACC is 100% and IV 0% (i.e. the sum is 1), which is roughly supported by the figure below. Over the full time period this gives a 22% contribution from IV and 78% from ACC. This is assuming an unrealistic instantaneous response and is given simply to illustrate that the way the authors have chosen to define the fractional attribution is, in my view, unhelpful.

Both reviewers raised the issue of this fractional metric. We've specifically chosen it because it is used extensively by the IPCC; and it was used by Marzeion et al (2014, M14), whose analysis we parallel. Moreover, the headline result of M14 was given in terms of this metric, and the result has been cited in subsequent IPCC reports and elsewhere in the literature. The word 'contribution' causes a strong reaction in some, especially when the number is negative or exceeds 100%, even if the equation hasn't changed. We note that the original M14 analysis allowed for a negative value in the uncertainty range and that, for anthropogenic warming, the current IPCC uncertainty range allows for it to exceed 100% of the observed. Apparently, there were divergent opinions in IPCC discussions over the use of word "contribution" and reports now avoid the word; and instead cast the assessment as the magnitude of anthropogenic warming relative to the observed. In our revisions we adopt this: we have removed the word contribution from the manuscript anywhere it might have this connotation. And we describe the analysis everywhere as the magnitude of the anthropogenic mass loss relative to the observed (or equivalently, relative to the total). In addition to these changes throughout the manuscript, when we introduce the metric, we now give readers two substantial paragraphs detailing the reasons for the metric, guiding them about how to interpret it, and now tell them ahead of time that we also analyze another metric in a later section. We hope that this sets a reader up better to interpret our results.

The two paragraphs are reproduced here: *"We here take a moment to discuss various possible metrics that can be used to characterize the influence of anthropogenic factors on glacier mass balance. Each have different merits. One approach is to set up a null hypothesis of a counterfactual, purely natural climate with no anthropogenic forcing, and to pose the question "how likely is the observed change in the natural case?". If the likelihood of the observed change falls below some stated level of significance, then the null hypothesis is rejected, and a "signal" can be declared detected. This was the approach applied to glacial retreat in Roe et al., 2017, and we apply it to mass balance in Section 4.4 of this paper. Another approach is to define a measure of the variance explained among an ensemble of climate models where natural variability, different anthropogenic forcing scenarios, and different models all contribute to the spread (e.g., Hawkins and Sutton, 2009; Marzeion et al., 2020), which is helpful for partitioning sources of uncertainty. A third approach, one that has been used by the Intergovernmental Panel on Climate Change (IPCC) over several past reports, is to estimate the magnitude of the anthropogenic component relative to the observed change. For instance, as of the 2018 IPCC Special Report, the central estimate of the magnitude of the anthropogenic change in global-mean surface temperature over the industrial era is that it is equal to 100% of the observed warming, with an assessed likely (i.e., a 2-in-3 chance) range of $\pm 20\%$ (Allen et al., 2018). Note this uncertainty range allows for the possibility that the magnitude of the anthropogenic warming exceeds 100% of the observed, because it is possible that the natural climate would have otherwise been cooling. M14 followed the IPCC in using this approach. It is useful, for instance, in the context of interpreting sea-level rise, where other contributions from thermal expansion and large ice sheets need to be understood. We want our results here to parallel those of M14, and we adopt the same approach as M14 in Sections 4.1 and 4.3 of this study.*

*M14 defined the following metric for comparing the magnitude of the anthropogenic mass loss to the FULL mass balance: $F_{anth} = 100\% \times (ANTH/FULL) = 100\% \times (1 - NAT/FULL)$. M14 calculated the 20-yr running mean of $F_{anth}$, in order to assess how anthropogenic mass loss has evolved over time at decadal scales. For assessing the total change in mass over the industrial era, ANTH and FULL can be integrated over time before calculating their ratio. Note that this cumulative ratio will not, in general, be equal to the time-average of $F_{anth}$. In our analyses, we present both the decadal and cumulative values. Both provide useful information: the 20-yr running mean provides insight into the causes of decadal-scale mass-balance observations, whereas the cumulative anthropogenic mass loss can be more directly tied to glacier retreat (see Section 4.4) and sea-level rise."*

Fig 8c suggests that the cumulative mb anomaly from 1850-1920 is ~ zero, for the case where =10 yrs, while the ACC % is 125% for the full period from 1850-2005, which is, again, counter intuitive.

It also doesn't seem to square with the integral of Fig 8e if the extreme values are ignored. In this case, the ACC % only exceeds (in a meaningful way) 100% from ~1960 onward. I do not understand, therefore, how a value of 125% is achieved, or in what way It is a meaningful way of representing the ACC contribution to mass change for this example.

The numbers work out. The NAT cumulative mass balance is small and positive, so because FULL = NAT+ANTH, the FULL and ANTH are similar in magnitude (& both negative). ANTH is slightly larger in magnitude, and therefore, a value for ANTH/FULL of 125% is correct.

We now note in both the text and the captions that the ratio of cumulative anthropogenic mass loss to total mass loss is not the same as the time-average of F_anth (since one can't average ratios in that way)

Likewise for the integral of 8h. For these shorter time constant glaciers, the ACC % only approaches 100% in the second half of the 20th C. This is not so far from the inference made in M14, at least for the short _glaciers.

See comments below, but even with its issues, the CESM reaches essentially 100% by 1950, which is a greater fractional value, and arrived at much earlier than M14. Again, see below, but we now give readers much clearer signposting about interpreting the CESM results.

For the longer time constant glaciers, presumably there is a +ve mb memory locked in that compensates for the >100% ACC percentage but this somehow contradicts the authors' _own claims in the abstract: "the anthropogenic component of the mass loss is essentially 100%." _In the case of large tau, the authors are stating it is 200%. If the authors stand by the values in Fig 8 then the ACC contribution for all glaciers is significantly >100% but nowhere is that stated or claimed for reasons that I believe the authors know themselves.

We don't stand by the values in Fig. 8! The CESM calculations should not be taken as the best assessment of the anthropogenic mass loss relative to the observed. The submitted manuscript already included several cautionary notes about the CESM model archive, but to guide a reader more clearly, the revised manuscript now groups them all into one paragraph, reproduced here:

*"It is important to note that the CESM ensemble has some limitations for our purpose here. First, over the industrial era, the model produces too little warming relative to observations, perhaps because it overestimates the response to aerosols (Otto-Bliesner et al., 2016). Secondly, we approximated the counterfactual, all-natural climate using model integrations with volcanic forcing only; and with so few ensemble members the ensemble mean still reflects a lot of unforced internal variability. Lastly, of course it is just one specific climate model, and other models might yield different sensitivities to natural and anthropogenic forcing. Our results with CESM should therefore not be considered a definitive calculation, but rather as just one scenario. The analysis does highlight the importance of using milennial-scale integrations for charaterizing preindustrial glacier variability (e.g., Huston et al., in press), and for correctly initializing preindustrial glacier states: more millennial-scale ensemble integrations from different modeling groups would be valuable. Specifically, there is a need for numerical experiments with all natural forcings included simultaneously, and with large enough ensemble members to confidently diasaggregate the forced and unforced climate response."*

The high percentages found in the CESM model should not be assumed correct, and moreover we believe they should be treated with a degree of skepticism, which we hope is now abundantly clear from the new paragraph. Our central estimate is based on assumptions that are stated in the discussion: that the central estimate of the anthropogenic warming is 100% of observed, and that the counterfactual, all-natural climate would have lain somewhere between a continued cooling and a reversion to the long-term average. Those assumptions are consistent with the published literature, but could be evaluated further (albeit in model world) with a larger suite of simulations using more GCMs. But we think the basis for the assessment is clearly stated, and a reader can disagree with the assumptions if they want to. An implication of our results is that while the central estimate might be 100%, the PDF of uncertainty is skewed towards larger percentages for larger tau glaciers. This is noted in the discussion.

In many respects, I found Fig 9 a more informative and clear demonstration of the role of ACC in post industrial glacier mb trends alongside the sentence starting at line 385.

As noted above, we now discuss various attribution metrics early in the paper, and directly point readers to this statistical framing early in the manuscript. While the analysis presented in Fig.9 has the form of a classical statistical test (evaluating a null hypothesis), which is a strength, it does not directly address the magnitude of the anthropogenic mass loss.

**Technical comments**
In the synthetic temperature figures (e.g. Fig 1) it wold be helpful to include a vertical dashed line in the length and mb columns indicating the start of the perturbation in temperature.

We've added this for Figs. 1 to 5. We made the lines light to try to avoid clutter, but hopefully it works.

Eqn 1. Replace = with ≈.

This is perhaps a small detail, but as it is a formula from the model equation already provided, we prefer to retain the equality symbol.

 This statement is misleading. The synthetic temperature experiments are useful to illustrate a point but they are **not** representative of the true temperature anomalies over the last 150 years and, in particular, the gradient of the ACC temperature trend over that time period. See Figs 1a and c above. There is a change in gradient from ~1960 onward. Something that is sort of apparent in Fig 9a and sort of implicitly captured in Figs 8e and h but not discussed at all.

We now note that the ACC temperature trend is not linear, but the point about anthropogenic mass loss does still hold. Our chosen, constant linear trend splits the difference between a slower early industrial warming and a more rapid later warming, but in both cases, it is more rapid than the prior natural cooling, meaning the anthropogenic mass loss still dominates across almost all of the industrial era. This is illustrated in the attached example figure, where the (synthetic) rate of warming more-than-quadruples in 1950.

[Figure]

Figure. Labels etc., are as for the main manuscript. A synthetic climate history with a rough quadrupling in the rate of warming in the mid 20th century.

L333 "The models" _=> The model.
Thanks!

L373 Fig 7e to I => Fig 9 e to i
Thanks!

L418 See previous comment. The definition used means it is > 100%, and for long     _glaciers closer to 200%.

We've changed the text to omit the word contribution. See above comments, but we don't rely on the CESM results (for the stated reasons) as the basis for our assessment – the answer depends on the assumptions about the counterfactual natural climate history, for which no single model should be trusted. The paragraph immediately following this sentence lays out the basis for our assessment. Even for long tau glaciers, the anthropogenic mass loss might be less than 100% of observed, but the uncertainty distribution skews towards higher values. This is stated clearly, and is now also referred to in the abstract.

There is a further inconsistency in the way this study has undertaken the difficult attribution part of the problem. As far as I can tell, the decline in length and mb in Fig 6 begins in ~1820, which is consistent with some temperature reconstructions for the last 2 millennia, which have an increase starting ~1800 of ~0.2 degs. That pre-dates any ACC contribution unlike the expt shown in Fig 5 and is not apparent in any other plots because they all start in 1850.

[Figure]

Figure. Same as Fig 6 in
the manuscript, but focusing on the interval 1750 to 2020

We're not sure what the inconsistency referred to is. Attached is a version of Fig. 6 zoomed in to show 1750 to 2020. If anything there is a slight cooling in the early 19th century (likely from Tambora and a

couple of others around that time); the onset of length retreat depends on the glacier response time; and the modeled mass balance is generally positive in the first half of the nineteenth century. But it doesn't really make a difference to our attribution analysis. Somewhere between 1850 and 1880 is generally taken to be the onset of significant anthropogenic forcing, and we focus on mass loss since 1850 so as to target the same interval as M14. The details of exactly when this occurs don't really matter for the industrial-era attribution. That is really the point here, the details of the preindustrial reconstructions don't strongly affect the central conclusions, since the rate of warming over the industrial era is so much greater. In the supplementary material of the submitted manuscript (SM, Fig. S3,S4) we considered the case of a step-function cooling from 1800 to 1850 in order to see if we could emulate the mass-balance deficit in 1880 presented in M14. If one instead supposed a step-function warming in the early 19[th] century it would throw the glaciers into positive mass balance once that warming ended.